# Influence of Sea Ice Anomalies on Antarctic Precipitation Using Source Attribution in the Community Earth System Model

Hailong Wang[1*], Jeremy G. Fyke[2,3], Jan T. M. Lenaerts[4], Jesse M. Nusbaumer[5,6,8], Hansi Singh[1], David Noone[7], Philip J. Rasch[1], and Rudong Zhang[1]

(1) Pacific Northwest National Laboratory, Richland, WA

(2) Los Alamos National Laboratory, Los Alamos, NM

(3) Associated Engineering, Vernon, British Columbia, Canada

(4) Department of Atmospheric and Oceanic Sciences, University of Colorado at Boulder, Boulder, CO

(5) NASA Goddard Institute for Space Studies, New York, NY

(6) Center for Climate Systems Research, Columbia University, New York, NY

(7) Oregon State University, Corvallis, OR

(8) Now at National Center for Atmospheric Research, Boulder, CO

[*]Correspondence to: Hailong.Wang@pnnl.gov

**Abstract**

We conduct sensitivity experiments using a general circulation model that has an explicit water source tagging capability forced by prescribed composites of pre-industrial sea ice concentrations (SIC) and corresponding sea surface temperatures (SST) to understand the impact of sea ice anomalies on regional evaporation, moisture transport, and source–receptor relationships for Antarctic precipitation in the absence of anthropogenic forcing. Surface sensible heat fluxes, evaporation, and column-integrated water vapor are larger over Southern Ocean areas with lower SIC. Changes in Antarctic precipitation and its source attribution with SICs have a strong spatial variability. Among the tagged source regions, the Southern Ocean (south of 50°S) contributes the most (40%) to the Antarctic total precipitation, followed by more northerly ocean basins, most notably the South Pacific Ocean (27%), South Indian Ocean (16%) and South Atlantic Ocean (11%). Comparing two experiments prescribed with high and low pre-industrial SIC, respectively, the annual mean Antarctic precipitation is about 150 Gt year$^{-1}$ (or 6%) more in the lower SIC case than in the higher SIC case. This difference is larger than the model-simulated interannual variability of Antarctic precipitation (99 Gt year$^{-1}$). The contrast in contribution from the Southern Ocean, 102 Gt year$^{-1}$, is even more significant, compared to the interannual variability of 35 Gt year$^{-1}$ in Antarctic precipitation that originates from the Southern Ocean. The horizontal transport pathways from individual vapor source regions to Antarctica are largely determined by large-scale atmospheric circulation patterns. Vapor from lower latitude source regions takes elevated pathways to Antarctica. In contrast, vapor from the Southern Ocean moves southward within the lower troposphere to the Antarctic continent along moist isentropes that are largely shaped by local ambient conditions and coastal topography. This study also highlights the importance of atmospheric dynamics in affecting the thermodynamic impact of sea ice anomalies associated with natural variability on Antarctic precipitation. Our analyses of the seasonal contrast in changes of basin-scale evaporation, moisture flux and precipitation suggest that the impact of SIC anomalies on regional Antarctic precipitation depends on dynamic changes that arise from SIC/SST perturbations along with internal variability. The latter appears to have a more significant effect on the moisture transport in austral winter than in summer.

## 1. Introduction

Antarctic surface mass balance (SMB), which plays a critical role in determining the evolution of the Antarctic Ice Sheet (AIS), controls the positive mass component of the overall AIS mass balance through precipitation (e.g., Lenaerts et al., 2012; Shepherd et al., 2012). Variations of AIS SMB, dominated by changes in precipitation (and to a lesser degree by sublimation), have implications for global mean sea level change. Modeling and experimental evidence suggests that AIS SMB increases in a warming climate due to increased precipitation as snowfall (e.g., Frieler et al., 2015; Zwally et al., 2015; Grieger et al., 2016; Lenaerts et al., 2016; Medley and Thomas, 2019). Previous studies have also attempted to attribute the increase in Antarctic moisture flux and precipitation to both thermodynamics (i.e., the increase in atmospheric moisture content) and dynamics (i.e., changes in the atmospheric circulation). Krinner et al. (2014) showed that changes in circulation patterns have a significant impact on Antarctic precipitation, but thermodynamic changes associated with ocean warming play a more important role in the projected increase in Antarctic precipitation. Grieger et al. (2016) quantified the thermodynamical and dynamical contributions to the increase of moisture flux and Antarctic precipitation by climate change projected in a multimodel ensemble and showed a decrease in dynamical contribution.

Observations and modeling have also shown strongly heterogeneous spatial patterns and temporal variability in AIS SMB and its trends (e.g., Thomas et al., 2017; Lenaerts et al., 2018; Medley and Thomas, 2019), suggesting the presence of regional precipitation variability over the AIS, which has been confirmed by previous studies using reanalysis and observational data (e.g., Bromwich et al., 2011; Behrangi et al., 2016; Palerme et al., 2017). Because of the extremely low atmospheric moisture content and low local moisture flux from the ice sheet surface, the formation of precipitation over Antarctica relies on moisture transport from the surrounding oceans (e.g., Tietäväinen and Vihma, 2008). By analyzing long quasi-equilibrium global climate model simulations, Fyke et al. (2017) identified statistically significant relationships in Antarctic basin-scale precipitation patterns that are driven by internal variability in large-scale atmospheric moisture transport. Sodemann and Stohl (2009) showed that the source regions for Antarctic precipitation over the Southern Ocean (SO) vary greatly between the ocean basins. Based on reanalysis datasets, Papritz et al. (2014) found that extratropical cyclones and fronts are key to the spatial distribution of evaporation and precipitation over the SO as well as moisture fluxes toward Antarctica. The impact of sea ice anomalies in the SO associated with internal variability on Antarctic moisture source apportionment as well as their feedback on atmospheric circulation remain unclear.

Sea ice has long been recognized as being highly sensitive to both forced changes and internal variability. Much of the SO is seasonally covered by sea ice. Oceanic areas close to the Antarctic coast are ice-covered most of the year, but the sea ice pack can be broken up by strong winds originating from the ice sheet, generating coastal polynyas that expose open ocean to the atmosphere. Variations in sea ice cover and/or the polynyas not only affect local surface heat and moisture fluxes from the ocean (e.g., Weijer et al., 2017) but also shift the latitudes of the mid-latitude storm track (e.g., Kidston et al., 2011). In contrast to the Arctic sea ice loss observed in recent decades, sea ice cover in the Antarctic (Southern Ocean) has increased over the last few decades (Turner and Overland, 2009), followed by a strong decline from 2016 (https://nsidc.org/data). While many coupled climate models are able to reproduce Arctic sea ice trends, these same models have difficulty simulating observed trends in sea ice cover over the Southern Ocean (e.g., Holland and Raphael, 2006; Meehl et al., 2016; Turner et al., 2013a). It is still unclear whether this trend in the Southern Ocean is due to internal climate variability, but there is no convincing mechanistic explanation for such responses of the SO sea ice cover to the warming caused by anthropogenic forcing. Given the connections between sea ice and Antarctic precipitation, this suggests a corresponding uncertainty in the projection of precipitation changes over Antarctica (Agosta et al., 2015; Bracegirdle et al., 2015) and, by consequence, AIS SMB and global sea level rise. Understanding the impact of sea ice anomalies on AIS SMB therefore presents an important scientific challenge (e.g., Kennicutt et al., 2015).

The direct impact of sea ice anomalies on moisture flux and Antarctic precipitation is through air-sea interactions, but the associated feedback on atmospheric dynamics can also be significant, as shown in previous modeling studies of projected climate change (e.g., Menéndez et al.,1999; Bader et al., 2013). Kittel et al. (2018) conducted sensitivity experiments in a regional climate model, with atmospheric circulation nudged toward reanalysis, to study the impact of idealized or forced sea ice perturbations on AIS SMB. They found significant Antarctic precipitation and SMB anomalies for the largest perturbations. However, the impact of SO sea ice anomalies and accompanying sea surface temperature (SST) changes on Antarctic snowfall changes through changing atmospheric moisture sources and associated atmospheric circulation and moisture transport, in the absence of anthropogenic forcing that primarily originates from low and mid-latitudes, has not been clearly disaggregated.

Moisture contributions from different source regions to local Antarctic precipitation cannot be quantified from direct measurements. Indirect approaches have to be used to derive such source–receptor relationships, characterize moisture history, and identify precipitation origins. Air parcel back-trajectory approaches tend to attribute more vapor sources to the high-latitude regions in the Southern Ocean (e.g., Helsen et al., 2007), likely due to the use of relatively short backward trajectories, which cannot trace water vapor originating from the distant low latitudes. A longer tracking time (e.g., 20 days) allows for

the identification of more distant moisture sources of Antarctic precipitation that are generally consistent with isotope-based source reconstructions and general circulation model (GCM) results (Sodemann and Stohl, 2009). However, for tracking times beyond 10 days, the single trajectory calculation error can become large due to the reduced coherency of air parcels (Sodemann et al., 2008), which might be

overcome stochastically by calculating many trajectories (Sodemann and Stohl, 2009). Despite their limitations (e.g., coarse resolution, numerical diffusion and biases in physics), atmospheric GCMs with moisture tracking capability using water isotope or tagged water tracers provide a powerful means to determine the origin of moisture sources of precipitation over receptor regions such as Antarctica (e.g., Koster et al., 1986; Delaygue et al., 2000; Noone and Simmonds, 2002; Singh et al., 2016a). These back-

trajectory and water tracer studies have shown that moisture sources for precipitation over the AIS are primarily from the Southern Ocean (south of 50°S) and the Southern Hemisphere mid-latitude oceans. Both back-trajectory and GCM water tracer approaches, along with ice core records of water isotopic composition, have been used to attribute water sources at Antarctic ice core sites and study their historical changes (e.g., Masson- Delmotte et al., 2011; Wang et al., 2013; Buizert et al., 2018).

In this study, we aim to understand the impact of SO sea ice anomalies associated with internal variability (in the absence of anthropogenic forcing) on local evaporation, moisture transport and source–receptor relationships for moisture and precipitation over Antarctica using a GCM that has an explicit water source tagging capability. Section 2 describes the GCM with water tagging capability and the experimental design. Main results and related discussions are presented in Section 3. Section 4 summarizes key

conclusions drawn from these sensitivity experiments and water source attribution analysis.

## 2.   Methodology

### 2.1 Model description

The climate model employed in this study is a coupled atmosphere-land version of the Community Earth System Model (CESM1-CAM5, CESM hereafter; Hurrell et al., 2013) that has an atmospheric water

tagging capability. This modeling tool has been used in several recent studies to quantify source-receptor relationships for the aerial hydrologic cycle (e.g., Singh et al., 2016a; Singh et al., 2016b; Singh et al., 2017; Nusbaumer and Noone, 2018; Tabor et al., 2018). We ran the atmospheric component of CESM, called the Community Atmosphere Model version 5 (CAM5), with prescribed sea surface temperature (SST) and sea ice concentrations (SICs) coupled with an interactive land component (CLM4, Oleson et

al., 2010), which includes the evolution of ice and snow over land. Snow cover over sea ice still evolves in the model, although SSTs and SICs are prescribed. CAM5 has relatively comprehensive

representations of surface evaporation, clouds, precipitation, and atmospheric circulation (Neale et al., 2010).

The atmospheric water tagging capability in CAM5 can be used to track water that enters the atmosphere through surface evaporation in any given region, moves with the air mass, condenses into liquid or ice

clouds, and forms precipitation (rain or snow). A set of new water variables (designated as a tagged water tracer set) is defined in CAM5 to capture the mass mixing ratio of vapor, cloud liquid, cloud ice, stratiform rain, stratiform snow, convective rain, and convective snow for each water source region of interest. Each water tracer set undergoes the same atmospheric processes as the corresponding standard water variables in the model. The tracked water cycle starts with surface evaporation/sublimation and

ends when water returns to the Earth's surface in the form of condensate or precipitation. Thus, the destiny of the tracer water is lost once it returns to the surface.

## 2.2 Experimental design

We use the water tagging capability along with a set of sensitivity experiments to examine the impact of changes in sea ice concentration (SIC) in the Southern Ocean on moisture transport, Antarctic snowfall,

and the AIS SMB. Here SIC is defined as the fractional area of ocean in a model grid that is covered by sea ice. Three SIC (and corresponding SST) composites are constructed from the pre-industrial control simulation of the CESM Large Ensemble (hereafter CESM LENS; Kay et al., 2015), which was initialized with January mean present-day potential temperature and salinity from the Polar Science Center Hydrographic Climatology dataset for the ocean and a previous CESM 1850 control run for the

atmosphere, land and sea ice. The CESM LENS control simulation was run for 1500 years with years 400-1500 released, which provides a continuous time series of over 1000 years to perform our composite analysis of SIC and SST based on monthly mean model output. A baseline simulation uses the mean SIC/SST distributions and two sensitivity simulations use the 10% lowest and highest annual average total Southern Hemisphere SIC, respectively, coupled to the corresponding anomalies in global SSTs. All

other forcing conditions (e.g., solar, greenhouse gases, anthropogenic aerosols) are identical across simulations. Although these sensitivity simulations are not designed to represent present-day conditions, several essential model fields from the baseline simulation are compared to the fifth generation ECMWF reanalysis (ERA5, 1979-2018). The main purpose is to provide a context for the interpretation of model results that might also be valid for the recent historical period in terms of internal climate variability. The

large-scale patterns of SIC, surface temperature, circulation (sea level pressure), precipitation, precipitable water, and horizontal moisture fluxes in the baseline simulation are comparable to those in

the ERA5 reanalysis, as shown in Figs. S1-S3, with differences in the circulation and moisture flux fields plotted in Fig. S4.

The three simulations (hereafter referred to as "mean", "low" and "high" according to the prescribed SICs) are conducted at a horizontal grid spacing of 0.9° × 1.25° with 30 vertical levels for 11 years. Results from the last 10 years are analyzed, assuming that the first simulation year is for model spin up. Figure 1 shows the anomalies of the two SIC composites with respect to the annual and seasonal (DJF and JJA) mean SIC. The most widespread SIC anomalies are found in the Weddell sea and the Bellingshausen and Amundsen seas in austral summer (DJF). The largest reduction in the "low" SIC case is along the east coast of the Antarctic Peninsula in DJF, while the most positive anomalies in the "high" SIC case are in the Amundsen sea away from the coastal zone in JJA. SIC anomalies are relatively small in the eastern Antarctic/SO sectors where the mean sea ice extent and SIC are also smaller. The regional difference in SIC anomalies adapted from the CESM LENS simulations is likely related to the key role of the Amundsen-Bellingshausen Seas Low (ABSL) in controlling the regional climate variability (e.g., Hosking et al., 2013). Although the magnitude and location of prescribed SIC anomalies are comparable to the observed SIC changes during recent decades (Hobbs et al., 2016), the prescribed seasonal SIC anomalies associated with internal variability under the CESM LENS pre-industrial conditions are likely to be different from future changes. Here the widespread anomalies occur in austral summer (DJF) and JJA anomalies concentrate at sea ice edges, while sea ice reductions by the end of the 21$^{st}$ century or in response to $CO_2$-doubling and the resulting global warming are expected to be dominated by winter (JJA) changes (e.g., Singh et al., 2017). Therefore, the simulations designed here are to examine Antarctic precipitation changes and moisture transport pathways dominated by natural variability, as opposed to the projected future changes driven by the increase in atmospheric moisture content related to temperature increases (e.g. Krinner et al., 2014, Frieler et al, 2015).

To use the water tracer tagging capability of CESM, we need to predefine water vapor source regions, where surface evaporation/sublimation of water provides the initial source of water vapor entering the atmospheric hydrologic cycle for the corresponding source region tags (Table S1). Figure 2 shows the water source regions, including major tropical, subtropical and mid-latitude ocean basins, land (all continents) and several finer sectors in the SO, that are tagged in all three simulations. According to Singh et al. (2017), the more distant lower-latitude oceans (i.e., 30°S equatorward) are much less efficient in contributing to Antarctic precipitation, and there is no seasonal sea ice over in the lower-latitude oceans, so each of these tagged regions is set up to cover a quite large area to economize computing time. Much finer divisions are used for the S. Ocean tags because they are in close proximity to the Antarctic and their surface evaporation is more affected by SIC variations. Five regular latitude-longitude boxes are defined.

The remaining area (irregular shape) of the SO was constructed by subtracting the sum of the five regular regions from the entire S. Ocean tag.

### 3. Results and Discussions

### 3.1 Responses of surface fluxes, precipitable water and precipitation to the SIC and SST anomalies

Although the three SIC composites were based on annual mean sea ice data, there are also large and consistent seasonal differences in SIC prescribed in the "low" and "high" sea ice cases (Fig. 1). The most widespread SIC differences are in the Weddell Sea and the King Haakon VII Sea where the reduction in "low" SIC extends to north of 60°S, while the largest difference (over 20%) occurs in the Bellingshausen and Amundsen Seas (Fig. 3a), indicating the role of the ABSL in dominating the overall internal

variability of sea ice cover in the Southern Ocean (e.g., Hosking et al., 2013). Compared to the "high" SIC case, the "low" SIC case also has much warmer SSTs and higher surface sensible heat flux and evaporation over the areas where SIC is lower (Figs. 3b, d and e). The sensible heat flux and evaporation over the northern latitudes of the SO also show large differences between the two cases due to meteorological responses (e.g., changes in winds as shown in Fig. S5; also changes in temperature and

specific humidity) to the SIC/SST differences. The total precipitable water (PW) in the low SIC case is greater over most of the SO, while the precipitation is greater over most of the coastal areas except for the King Haakon VII Sea (Fig. 3c and f). Comparison to the corresponding decadal variability of these annual mean fields (Fig. S6), along with a Student's t-test at 90% confidence, suggests that the significant regional differences in surface temperature, evaporation and precipitable water are mostly due to SIC/SST

perturbations while changes in precipitation are influenced more by internal variability.

There are seasonal contrasts between DJF and JJA that can be an indication of the relative importance of SIC anomalies and internal variability. As shown in Figs. S7 and S9, SIC differences in DJF are more widespread (e.g., large SIC changes near coastal areas) than in JJA when SIC changes are concentrated at the sea ice edge. The differences in surface temperature and heat fluxes within the sea ice zone is much

larger and more definite in JJA than in DJF, which is similar for the seasonal contrast in SLP over SO and Antarctica (Fig. 4). However, the decadal variability of these fields is also stronger in JJA than in DJF (See Figs. S8, S10 and S11). It indicates that decadal variability plays a more important role in determining the moisture flux and precipitation differences in JJA than in DJF. Comparing the regional changes in seasonal evaporation and precipitation (Figs. S7 and S9), positive evaporation anomalies in the

SO can only translate to a positive impact on Antarctic basin-scale precipitation when there is a strong meridional moisture flux towards the basin (Fig. S2). This is consistent with the finding of Fyke et al. (2017) that large-scale moisture transport is the main driver of basin-scale precipitation variations over

Antarctica. For example, evaporation anomalies are significant and positive in both DJF and JJA over the Amundson Sea, but the meridional flux ($F_{VQ}$) is much stronger in JJA than in DJF, leading to a more significant positive impact on the downwind Antarctic coastal precipitation in JJA.

### 3.2 Changes in meridional transport and circulation patterns

SIC changes between the "low" and "high" cases can be closely related to large-scale circulation changes over the SO. Previous studies identified complex large-scale interactions between the atmosphere and Antarctic sea ice cover that depend on the geographic location of sub-sectors in the SO (e.g., Lefebvre and Goose, 2008; Hobbs et al., 2016). Meridional winds can drive the exchange of dry/cold air over the AIS with moist/warm air from lower-latitude oceans. In the annual mean, moisture from the north moves

to Antarctica over the entire SO, while southerly katabatic outflow brings relatively dry air back to the ocean. The meridional moisture flux ($F_{VQ}$) that is largely determined by meridional winds is also significantly different between the "low" and "high" SIC cases (Fig. 4a). Changes in meridional winds can be explained by the sea level pressure change using the geostrophic balance approximation (Fig. S5 and Fig. 4c). The pattern of variations in meridional moisture flux (Fig. 4a) is consistent with

precipitation differences (Fig. 3f). Decreases in precipitation in the "low" SIC case over the King Haakon VII Sea and coastal areas can be traced to the reduction in meridional flow and related moisture fluxes from the north in part due to the SIC decrease and internal variability (Fig. 4a). A Student's t-test and the comparison of changes to decadal variability (Fig. 4 and S11) suggest that the reduction of meridional moisture flux ($F_{VQ}$) in that area is primarily determined by the SIC decrease in JJA but more likely due to

internal variability in DJF. Therefore, the impact of sea ice anomalies and corresponding SST changes on Antarctic precipitation stem both from their thermodynamic impact on moisture sources and from the dynamic changes that accompany the different SIC and SST patterns as well as internal variability.

    Comparing the "low" and "high" cases also reveals a strengthening of the Hadley Cell and weakening of the polar vortex in the southern hemisphere accompanying the "low" SIC (figure not shown). Variations

in zonal flow and moisture fluxes over much of the SO (Fig. 4b and Fig. S5) can affect Antarctic precipitation through redistribution of moisture among the different sectors/basins and indirect changes in northward moisture transport. Regional westerlies can also drive changes in upper-ocean heat storage and sea ice formation by affecting Ekman pumping and thus the sea ice extent (e.g., Turner et al., 2013b). The southern annular mode, which dominates the variability of the large-scale atmospheric circulation in the

Southern Hemisphere, has been found to co-vary with tropical SST variability (e.g., Ding et al., 2012) and respond to SIC changes (e.g., Menéndez et al., 1999; Bader et al., 2013; Smith et al., 2017). The ABSL, which plays an important role in bringing warm/moist air into the Bellingshausen Sea and Antarctic

Peninsula region and moving cold/dry air equatorward through the Ross Sea region, strongly influences winds, near-surface temperature, precipitation and SIC over the Amundsen-Bellingshausen Seas (e.g., Hosking et al., 2013). Conversely, the strength and location of the ABSL (in JJA) can also be affected by the sea ice and temperature changes along with internal variability, as depicted in Fig. 4c. Therefore, variability in atmospheric circulation and SIC/SST anomalies indirectly influence moisture transport and regional precipitation over Antarctica. Here we cannot elaborate more on causes of CESM-simulated SO SIC/SST anomalies in the Large Ensemble that promulgate the resulting circulation changes when prescribed in our sensitivity experiments. To further separate the direct impact of changes in evaporation from the indirect impact of changes in circulation and moisture transport associated with SIC/SST anomalies as well as internal variability, a future dedicated study using a series of carefully designed experiments (e.g., with specified atmospheric circulations and/or regional evaporation) is needed.

### 3.3 Seasonal variation of Antarctic precipitation and source attribution

As expected, there are strong seasonal variations in total Antarctic precipitation, with a distinct minimum in austral summer months (Fig. 5), which is opposite to the PW seasonal cycle (Fig. S12). Although the seasonal pattern itself changes very little with the SIC/SST anomalies, the magnitude of seasonal precipitation has relatively larger changes, as well as larger interannual variability (indicated by the longer error bars), in spring and fall than the other months, which is consistent with SIC changes between the "low" and "high" cases (Fig. 1). The annual mean precipitation is about 150 Gt year$^{-1}$ more in the "low" SIC case than in the "high" SIC case, representing a 6% increase relative to the total precipitation (2500 Gt year$^{-1}$) in the "mean" SIC case. This difference is larger than the interannual variability of Antarctic precipitation (99 Gt year$^{-1}$) that is characterized by one standard deviation of annual mean precipitation within the 10 years of the "mean" SIC case. Note that the standard deviation of annual mean Antarctic precipitation for the entire CESM LENS time series is 98 Gt year$^{-1}$, which is smaller than the variability of 122 Gt year$^{-1}$ for recent historical precipitation simulated in CESM (Fyke et al., 2017). For reference interannual variability in Antarctic precipitation calculated from the ERA5 reanalysis (1979-2018) is 113 Gt year$^{-1}$. The contrast in Antarctic precipitation contributed by the S. Ocean between the "low" and "high" SIC cases, 102 Gt year$^{-1}$, is much larger than the interannual variability of 35 Gt year$^{-1}$ in precipitation that originates from the S. Ocean, although it is a small fraction of the increase in evaporation (870 Gt year$^{-1}$) from the S. Ocean (again comparing the "low" SIC case to the "high" SIC case).

Among the tagged source regions, the S. Ocean (including the 6 sub-sectors) contributes the most (40%) to the Antarctic total precipitation in the "mean" SIC case, followed by S. Pacific Ocean (27%), S. Indian

Ocean (16%) and S. Atlantic Ocean (11%), with the remaining mostly coming from evaporation/sublimation over land. The other oceans in the tropics and northern hemisphere have a negligible contribution to Antarctic moisture and precipitation. The fractional contribution by the S. Ocean has a 1.7% increase (comparing the "low" SIC case to the "high" SIC case), while there is a small decrease from the S. Atlantic (-0.7%) and S. Pacific (-1%). The contribution by the S. Ocean, Land and some remote oceans (e.g., S. Indian Ocean and S. Pacific Ocean) has a relatively strong seasonal variation. There is a seasonal peak contribution from the S. Ocean in fall and spring (MAM and SON), when the SIC anomalies make a relatively large difference to the total Antarctic precipitation (Fig. 5), while the peak is in boreal summer (JJA) for the remote oceans and in austral summer (DJF) for land sources. The annual mean contribution of 40% by the S. Ocean is larger than the estimate (30%) by Sodemann and Stohl (2009) using the 20-day back trajectory method for a specific historical time period (1999-2005). Also different from the finding of Sodemann and Stohl (2009), the seasonal cycle of the S. Ocean contribution to Antarctic precipitation in our study is not mainly determined by the SIC seasonality. These may be due in part to differences in SIC/SST conditions and atmospheric circulations (rather than the tools being used), especially for the JJA source attribution to evaporation over the Amundsen and Bellingshausen Seas, where the internal variability of relevant fields is large (Figs. S10 and S11).

As shown in Fig. (3f), the responses in Antarctic precipitation to SIC/SST anomalies along with internal variability have a strong spatial variability, as does the source attribution. Figure 6 shows the spatial distribution of fractional contributions to annual mean Antarctic precipitation by individual and combined source tags. The five major source regions together account for over 95% of total Antarctic precipitation, with individual regions dominating in certain areas as determined by geographical location and atmospheric circulation patterns (Fig. S3). The S. Ocean tag as a whole dominates precipitation over most of the coastal areas except for the segment (90–150°E) located at the south of the S. India Ocean.  The sub-sector sources in the SO primarily affect nearby coastal areas as well as downwind coastal and inland regions. There is also a strong regional variation in the annual and seasonal changes of absolute precipitation and corresponding fractional contribution from individual source regions related to the SIC anomalies (Figs. S13-S18). The higher fractional contribution in the lower SIC case from the S. Ocean and sub-sectors is mostly due to increased coastal precipitation, while changes in the fractional contribution by the remote sources do not correspond well with the absolute precipitation change over the SO and Antarctica. This arises because small increases in precipitation originating from remote sources can be overwhelmed by large increases from local sources. Such compensating effects occur not only between the local source region (S. Ocean) and remote source regions but also amongst the remote region

contributions themselves. Another reason is that the long-range moisture transport from remote source regions towards Antarctica is more likely affected by internal variability in atmospheric circulations. A Student's t-test suggests that S. Ocean has a more significant impact on the response of Antarctic precipitation to the SIC/SST anomalies than the remote oceans do. The total response from all major sources are less robust than S. Ocean alone for annual and seasonal mean results (i.e., comparing "Sum" to "S. Ocean" in Figs. S13-S18). Similarly, smaller source regions such as the sub-sectors of S. Ocean tend to impose more robust signals than the S. Ocean as a whole, indicating that the quantified response of Antarctic precipitation to SIC/SST anomalies in the S. Ocean subsectors has minimal interference from internal variability.

To further look at spatial variations in precipitation and its source attribution, we divide Antarctica into three broad sectors: eastern Antarctica (0, 180°E; 65°S, 80°S), western Antarctica (0, 180°W; 65°S, 80°S), and interior Antarctica (80°S, 90°S). The contribution of the entire S. Ocean source tag to the annual mean precipitation dominates over all three and has a small interannual variation, although seasonal variations of contribution have large differences (Fig. S19). The S. Ocean has a larger contribution to precipitation over western Antarctica than eastern Antarctica, which is due in part to higher elevation in the east. Subsectors of the S. Ocean in the west (e.g., Amundsen Sea, Weddell Sea, and part of Ross Sea) can have a discernable impact on precipitation over interior Antarctica (Fig. S19), which shows a significant response to SIC/SST anomalies in these source regions as well (Figs. S13-S18). Among the major remote source regions, the S. Indian Ocean and S. Atlantic dominate the contribution to precipitation over eastern Antarctica, while the S. Pacific Ocean dominates over western and interior Antarctica, especially in austral winter (JJA).

### 3.4 Transport pathways of water to Antarctica

As indicated in the previous section (Fig. 6), the horizontal transport pathways of atmospheric water from individual source regions to a receptor are largely determined by large-scale atmospheric circulations. Localized or large-scale vertical lifting at the source region or along the transport pathway is an important factor in determining the extent to which water vapor can penetrate to the Antarctic interior before precipitating. Stohl and Sodemann (2010) illustrated the thermodynamic transport and lifting barrier for SO low-level airmasses to move to the Antarctic interior. Figure 7 shows the vertical distribution of fractional contribution to zonal mean water vapor mixing ratio from the major source regions. In general, vapor originating from remote source regions at lower latitudes takes elevated pathways to Antarctica while vapor from the nearby sources in the SO moves southward within the lower troposphere, as also noted in previous studies (e.g., Noone and Simmonds, 2002; Sodemann and Stohl, 2009; Stohl and Sodemann, 2010; Kittel et al., 2018). The meridional and vertical transport of vapor is along zonal mean

moist isentropes ($\theta_e$) that are largely shaped by local airmass temperature and topography in Antarctica, especially, for water vapor originating from the individual SO sub-sectors (Fig. 8; see also Bailey et al 2019). As a result, a large portion (up to 70% for the zonal mean) of the vapor below 700 mb comes from the S. Ocean source tag, which also contributes a significant amount (10-40%) to vapor in the mid-troposphere (700–400 mb). Vapor in the upper troposphere (above 400 mb) predominantly comes from remote oceans through elevated pathways, although evaporation from lower-latitude continents also contribute a discernible fraction (up to 20%). Vapor originating from the equatorial oceans, lifted by deep convection in the ITCZ, can have a substantial contribution (up to 40%) at very high levels (above 200 mb).

We have shown in the previous section that the SO SIC reduction substantially increases the atmospheric column-integrated water vapor (Fig. 3). Vertical distributions of water vapor changes show that the increase occurs mostly in the lower troposphere over the SO and coastal areas (Fig. S20), where water vapor sources include the S. Pacific, S. Indian Ocean and S. Atlantic in addition to the primary contributor, the S. Ocean (Fig. 7). However, two of the three major remote ocean source regions (S. Pacific and S. Atlantic), Equatorial Oceans and land contribute significantly less water vapor further inland in the "low" SIC case (Fig. S20), which leads to a discernable and significant reduction in their fractional contribution to water vapor in the lower and mid troposphere (Fig. 9). The contribution by the entire S. Ocean tag increases substantially south of 50°S in the "low" SIC case, compensating for the reduced contribution from remote oceans. Note that the changes in fractional contribution in the upper troposphere and lower stratosphere (Fig. 9) are more likely related to SST and deep convection changes in the lower latitudes than to the SIC changes.

### 4.  Summary and Conclusions

In this study, we use the Community Atmosphere Model version 5 (CAM5) with explicit water tagging capability to quantify the impact of sea surface temperature (SST) and sea ice concentration (SIC) changes on the moisture sources of Antarctic precipitation. A set of sensitivity experiments are conducted to understand the impact of SIC and SST variations on regional evaporation, moisture transport, and source–receptor relationships for Antarctic precipitation. Three composites of sea ice concentration (SIC), which were constructed from the 1000-year fully-coupled pre-industrial control simulation of the CESM Large Ensemble Project using mean, 10% lowest, and 10% highest SIC years (and corresponding SSTs), respectively, are used as prescribed boundary conditions for 10-year atmosphere-only simulations. Moisture originating from individual geographical regions is explicitly tracked using separate water tracers throughout the atmospheric water cycle that closes with surface precipitation.

Because of the prescribed changes in the SIC and SST, surface sensible heat fluxes and evaporation over lower SIC areas in the SO increase significantly in the "low" SIC case, compared to the "high" SIC case, especially in JJA. Column-integrated water vapor also increases over much of the SO, while changes in Antarctic precipitation with SICs have a strong spatial variability, as does the source attribution. The prescribed SIC anomalies in DJF are more widespread than in JJA when SIC changes are concentrated at the sea ice edge. Our analysis indicates that decadal variability plays a more important role in determining the moisture flux and precipitation differences in JJA than in DJF. Comparing the regional changes in seasonal evaporation and precipitation, positive evaporation anomalies in the SO can only translate to a positive impact on Antarctic basin-scale precipitation when there is a strong meridional moisture flux towards the basin.

Among the tagged source regions, the S. Ocean (including all six sub-sectors) contributes the most (40%) to the Antarctic total precipitation, followed by the S. Pacific Ocean (27%), S. Indian Ocean (16%) and S. Atlantic Ocean (11%), with the remaining contributions mostly from evaporation or sublimation over global land. The major remote source regions have a reduced absolute contribution to water vapor further inland in the "low" SIC case, which leads to a significant reduction in their fractional contribution, especially, in the lower and mid troposphere. With lower SIC, the relative contribution to water vapor south of 50°S by the S. Ocean tag increases substantially, compensating the reduction in the relative contribution from remote oceans. This is qualitatively consistent with the annual mean source attribution change in response to warming from $CO_2$ doubling (Singh et al., 2017). The annual mean total Antarctic precipitation is approximately 150 Gt year$^{-1}$ more in the "low" SIC case than in the "high" SIC case. This difference is larger than the interannual variability of Antarctic precipitation (characterized by one standard deviation of annual mean precipitation) estimated from the CESM LENS control experiment and the ERA5 reanalysis (1979-2018), 98 and 113 Gt year$^{-1}$, respectively.  The contrast in precipitation between the "low" and "high" SIC cases contributed by the S. Ocean, 102 Gt year$^{-1}$, is even more significant, compared to the interannual variability of 35 Gt year$^{-1}$ in precipitation that originates from the S. Ocean.

The horizontal transport pathways from individual vapor source regions to Antarctica are largely determined by the large-scale atmospheric circulation, which confirms earlier findings (e.g., Stohl and Sodemann, 2010; Singh et al., 2017). Localized or large-scale vertical lifting is important in determining the heights at which vapor is transported and forms cloud. Thus the source contribution is primarily determined by their geographical location (and atmospheric dynamical setting) and atmospheric circulation patterns, as well as the local elevation over Antarctica. Vapor from source regions at lower latitudes takes elevated pathways to Antarctica while vapor from the nearby tags in the SO moves

southward within the lower troposphere. The entire S. Ocean source tag is the primary contributor to the annual mean precipitation over all defined Antarctic sub-regions - eastern Antarctica (0, 180°E; 65°S, 80°S), western Antarctica (0, 180°W; 65°S, 80°S), and interior Antarctica (80°S, 90°S). However, it has a larger contribution to precipitation over western Antarctica than eastern Antarctica, which is in part due

to higher elevation in the east. The S. Ocean contribution also has large seasonal differences among the three. Among the remote source regions, the S. Indian Ocean and S. Atlantic dominate the contribution to precipitation over eastern Antarctica, while the S. Pacific Ocean dominates over western and interior Antarctica, especially in austral winter (JJA).

In addition to direct thermodynamic effects, the impact of sea ice anomalies on regional precipitation over

Antarctica also depends on atmospheric circulation changes that result from the SIC/SST perturbations prescribed to the simulations along with internal variability. Regional anomalies in zonal and meridional winds combine with surface evaporation changes to determine regional shifts in zonal and meridional moisture fluxes. The resultant changes in meridional moisture fluxes from the Southern Ocean to the Antarctic continent can explain some of the precipitation differences between the "low" and "high" SIC

cases. Variations in zonal moisture fluxes can also affect Antarctic precipitation indirectly through the redistribution of moisture among the different sectors/basins. The seasonal contrast between DJF and JJA in basin-scale moisture flux and precipitation changes can be used as an indication of the relative importance of SIC anomalies versus internal variability. However, the experiment design of this study doesn't allow us to isolate the impact of SIC anomalies from internal variability on circulation-driven

changes in Antarctic precipitation. A future dedicated study with specified large-scale circulations or fixed regional evaporation might be helpful in this regard.

*Code and data availability.* The CESM model code can be obtained from http://www.cesm.ucar.edu/ and https://github.com/NCAR/iCESM1.2. Directions for obtaining CESM Large Ensemble data are available

at www.cesm.ucar.edu/projects/community-projects/LENS/. ERA5 reanalysis products were downloaded from the Climate Data Store https://cds.climate.copernicus.eu/cdsapp#!/dataset/reanalysis-era5-single-levels-monthly-means?tab=overview. The model simulations will be made available through https://datahub.pnnl.gov upon request to the corresponding author.

*Competing interests.* The authors declare that they have no conflict of interest.

*Author Contribution.* HW, JF, and JL designed the research. DN and JN provided technical support of running the CESM model code with water tagging. JF did the SIC and SST composites. HW conducted

the model simulations. HW and RZ did the analysis and made the figures. All authors contributed to the discussion of results and writing of the paper.

*Acknowledgments.* This research is based on work supported by the U.S. Department of Energy (DOE), Office of Science, Biological and Environmental Research as part of the Regional and Global Model Analysis (RGMA) program. Jan T. M. Lenaerts acknowledges support from the National Aeronautics and Space Administration (NASA) through project 80NSSC18K1025. Jesse Nusbaumer was supported by the NASA Post-doctoral Program (NPP) fellowship. The Pacific Northwest National Laboratory (PNNL) is operated for DOE by Battelle Memorial Institute under contract DE-AC05-76RLO1830. The CESM project is supported by the National Science Foundation and the DOE Office of Science. We would like to acknowledge high-performance computing support from Yellowstone (ark:/85065/d7wd3xhc) provided by NCAR's Computational and Information Systems Laboratory, sponsored by the National Science Foundation.

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

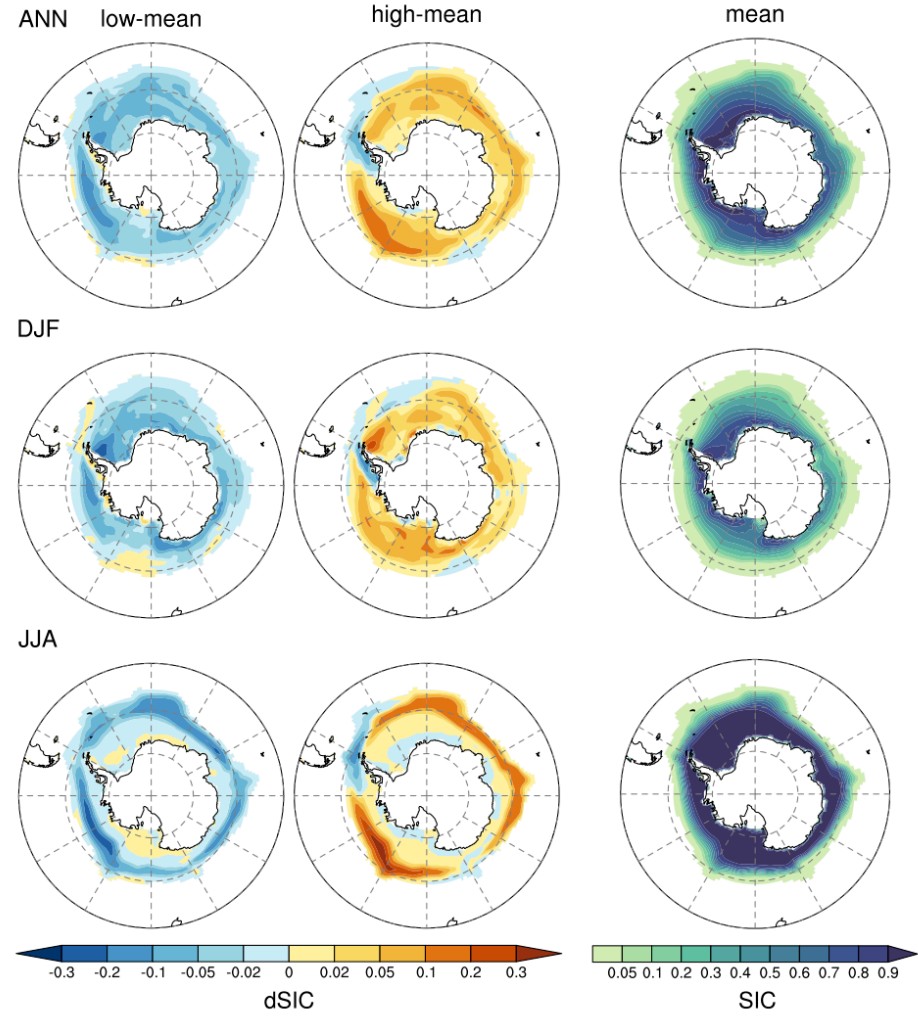

Figure 1: the anomalies of the two SIC composites ("low" and "high") with respect to the annual and seasonal mean SIC ("mean" in the right-most column).

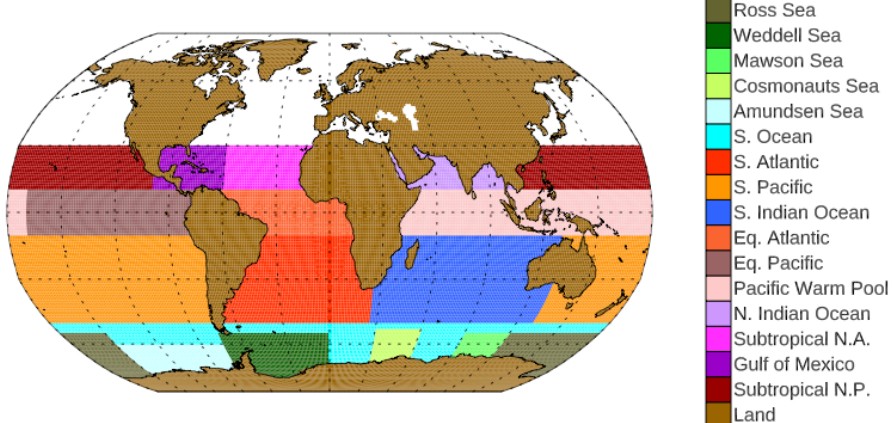

Figure 2: Tagged water source regions that are potentially important for Antarctic precipitation, including all major tropical/subtropical and mid-latitude ocean basins (Subtropical N. Pacific, Subtropical N. Atlantic, Gulf of Mexico, Pacific Warm Pool, Equatorial Pacific, Equatorial Atlantic, N. Indian Ocean, S. Indian Ocean, S. Pacific, S. Atlantic, and S. Ocean), five finer sectors (Amundsen Sea, Cosmonauts Sea, Mawson Sea, Weddell Sea, and Ross Sea) in the Southern Ocean, and land (all continents). All remaining oceanic areas (white) are also tagged.

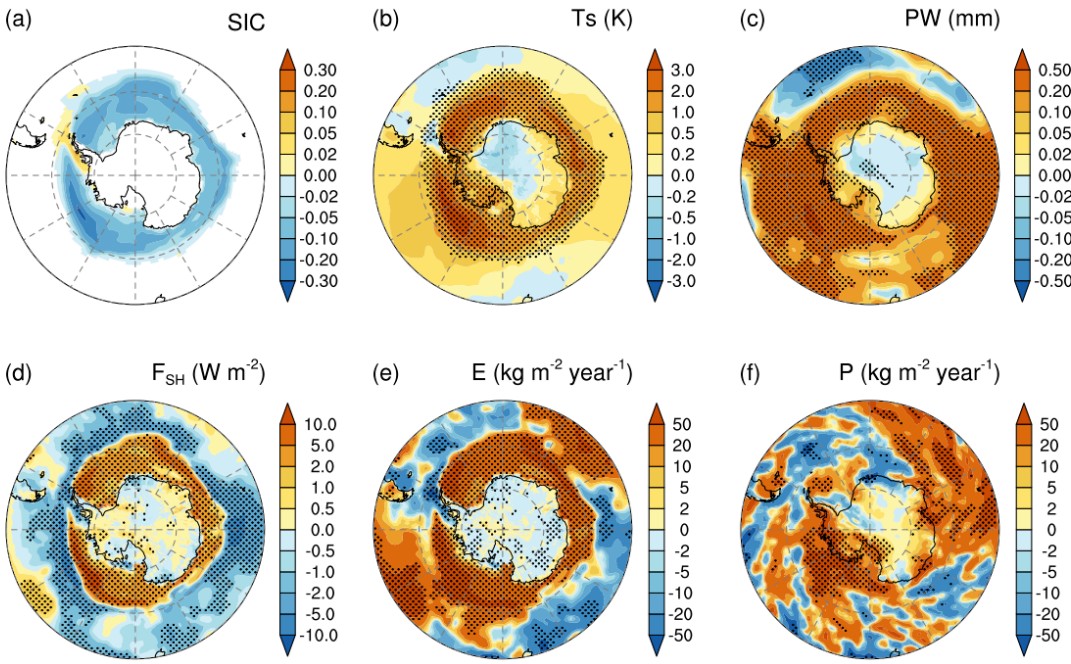

Figure 3: Annual mean differences in (a) sea ice concentrations (SIC), (b) surface temperature (Ts), (c) total precipitable water (PW), (d) surface sensible heat flux ($F_{sh}$), (e) surface evaporation/sublimation (E), and (f) surface precipitation (P) between the "low" and "high" SIC cases. Stippling on the maps indicates that the differences are statistically significant at the 90% confidence level based on Student's *t*-test.

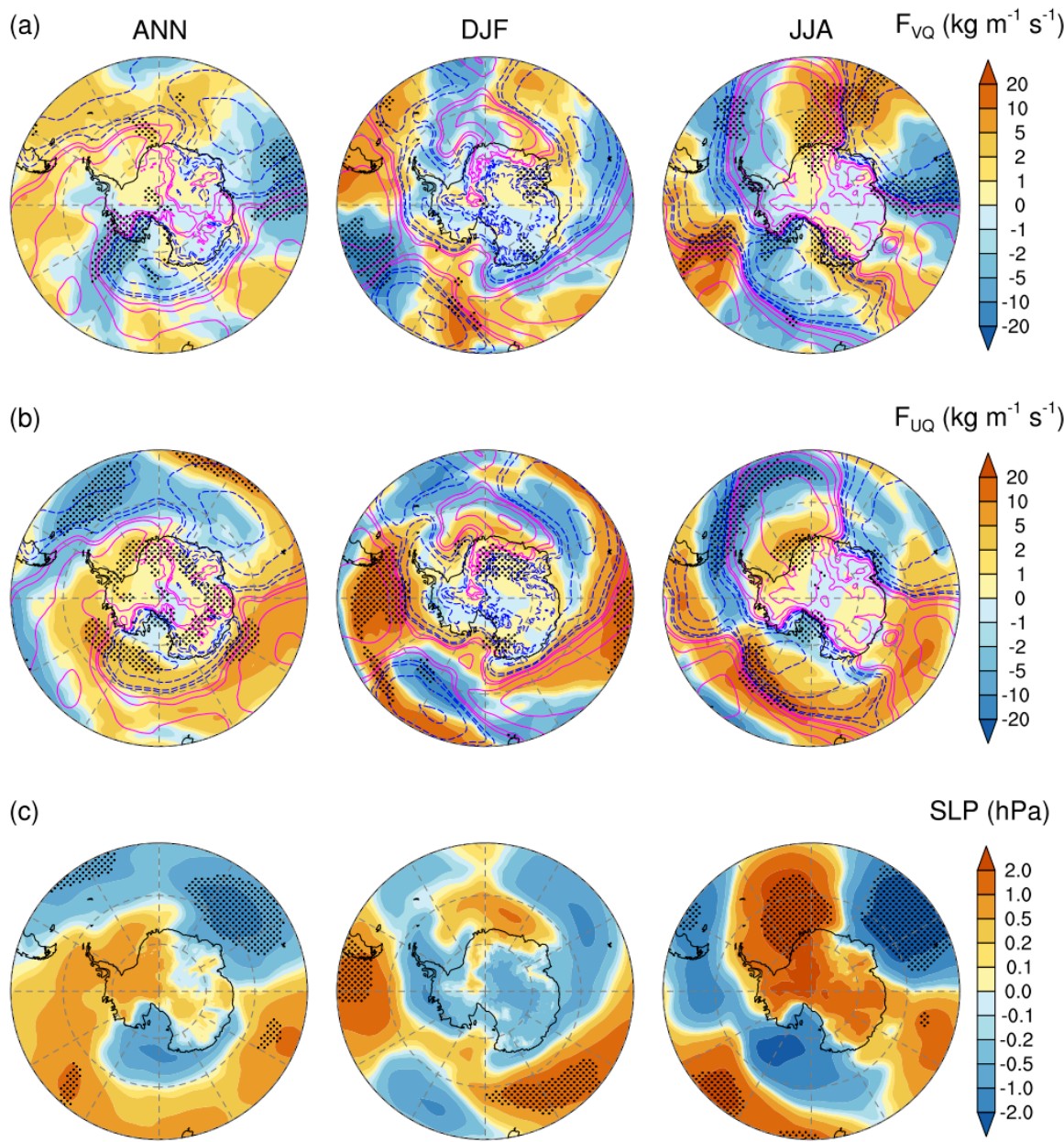

Figure 4: Spatial distribution of differences ("low" minus "high") in annual (left) and seasonal (DJF and JJA) mean column-integrated (a) meridional and (b) zonal moisture flux, and (c) sea level pressure (SLP). The superimposed contour lines represent SLP differences (magenta for positive and blue for negative with the same intervals as in the SLP color bar in hPa). Stippling on the maps indicates that the differences are statistically significant at the 90% confidence level based on Student's *t*-test.

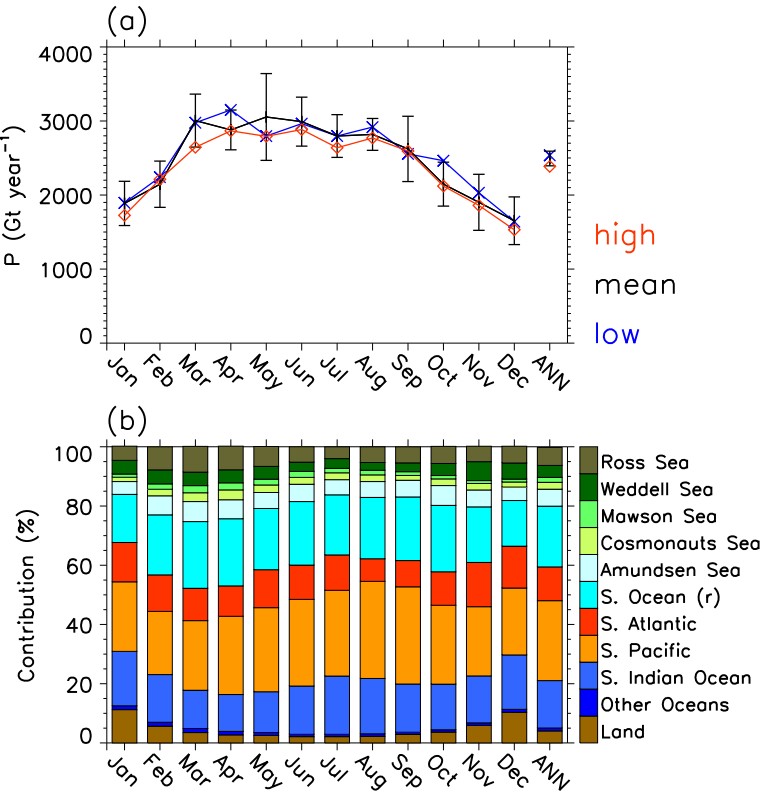

Figure 5: seasonal variation (January-December) and annual mean (ANN) precipitation over Antarctica in the three simulations (top) and the corresponding fractional contributions by the tagged source regions from the "mean" (bottom). Error bars represent one standard deviation of corresponding results from 10 individual years of the "mean" case. Note that the S. Ocean (r) tag plus the five sub-sector tags represent the entire Southern Ocean. Contributions from tropical oceans and northern hemisphere oceans are combined to the "Other Oceans".

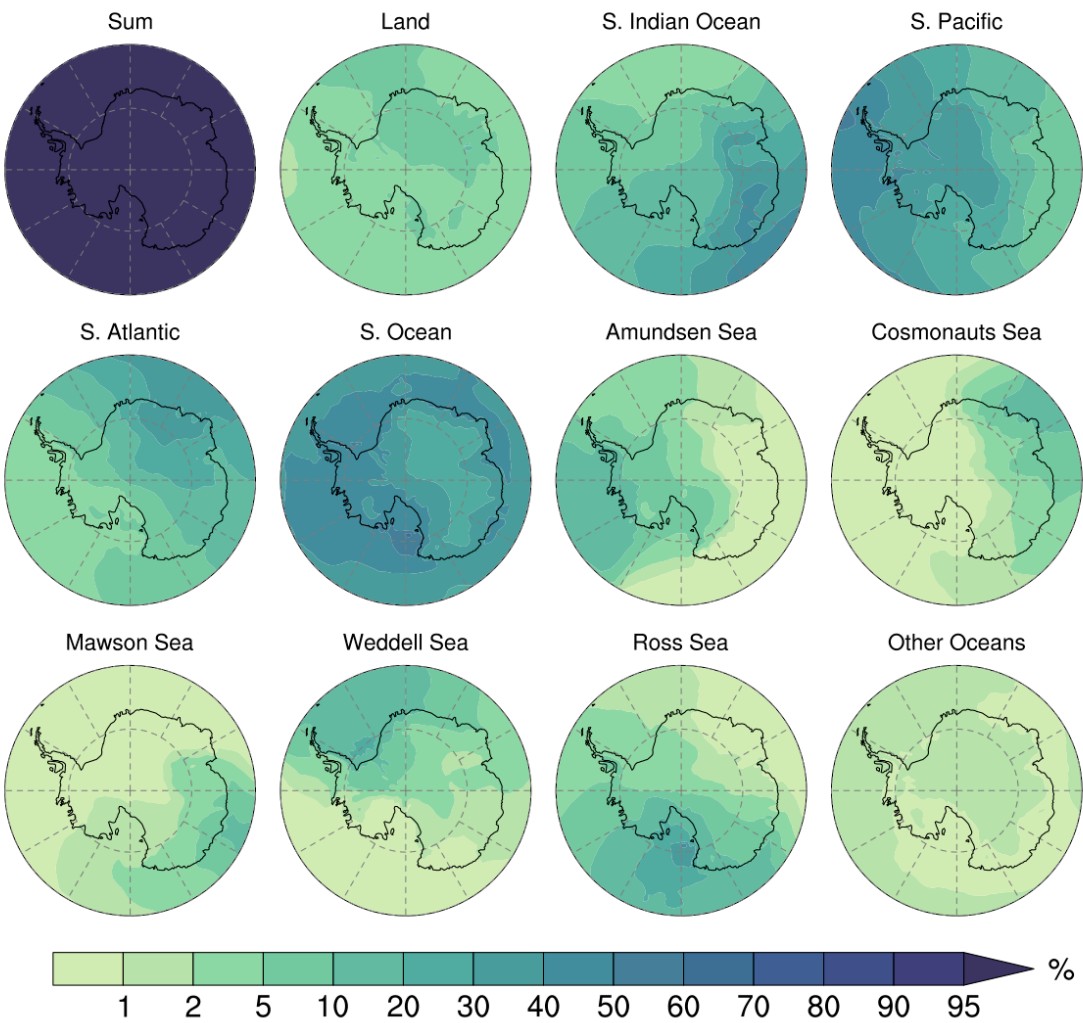

Figure 6: Spatial distribution of fractional contribution (%) to annual mean precipitation at the surface from individual source regions in the "mean" case. The "Sum" (upper-left panel) represents contributions from the five major source regions, including Land, S. Indian Ocean, S. Pacific, S. Atlantic and S. Ocean. Contributions from tropical oceans and northern hemisphere oceans are combined to the "Other Oceans".

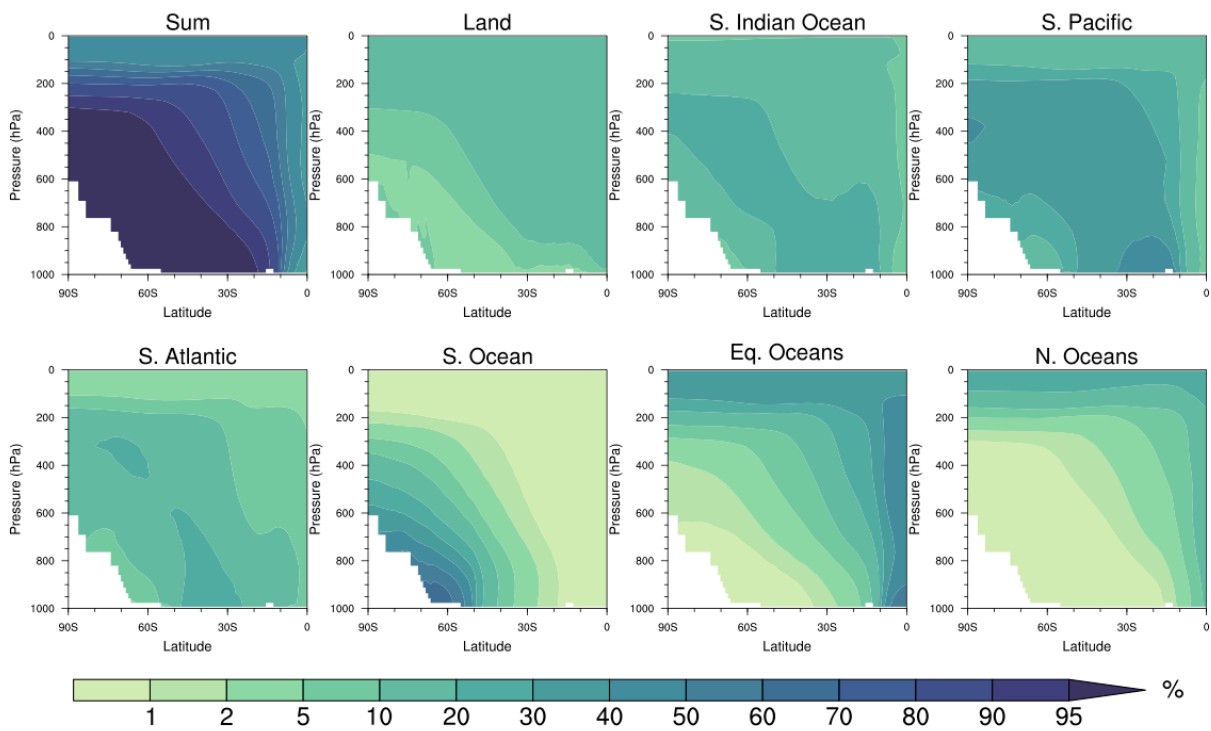

Figure 7: Vertical distribution of fractional contribution (%) to annual and zonal mean water vapor mixing ratio from individual source regions in the "mean" case. The "Sum" (upper-left panel) represents contributions from the five major source regions, including Land, S. Indian Ocean, S. Pacific, S. Atlantic and S. Ocean. The S. Ocean tag here includes all six sub-sectors. The Eq. Oceans includes the three equatorial ocean tags, and the N. Oceans includes the remaining ocean tags in the northern hemisphere.

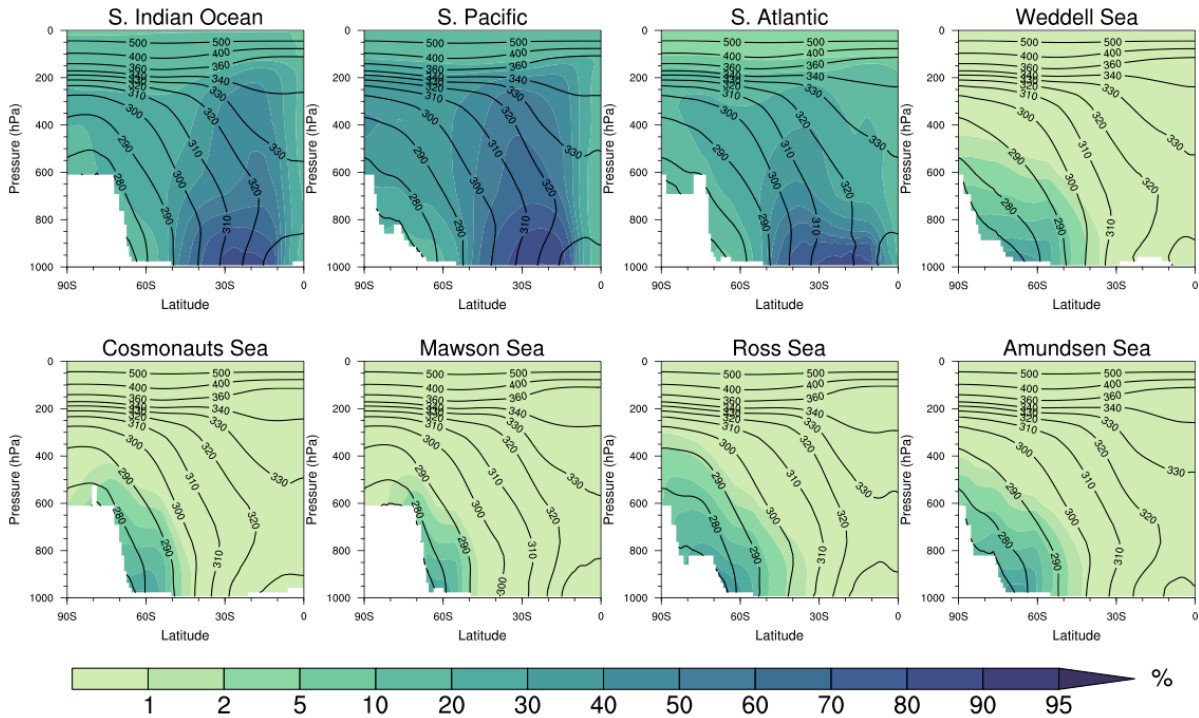

Figure 8: Vertical distribution of fractional contribution (%) to annual and zonal mean water vapor mixing ratio from individual source regions in the "mean" case. Contour lines are zonal mean equivalent potential temperature ($\theta_e$). Zonal mean in each panel is taken along the corresponding longitude band of the source region.

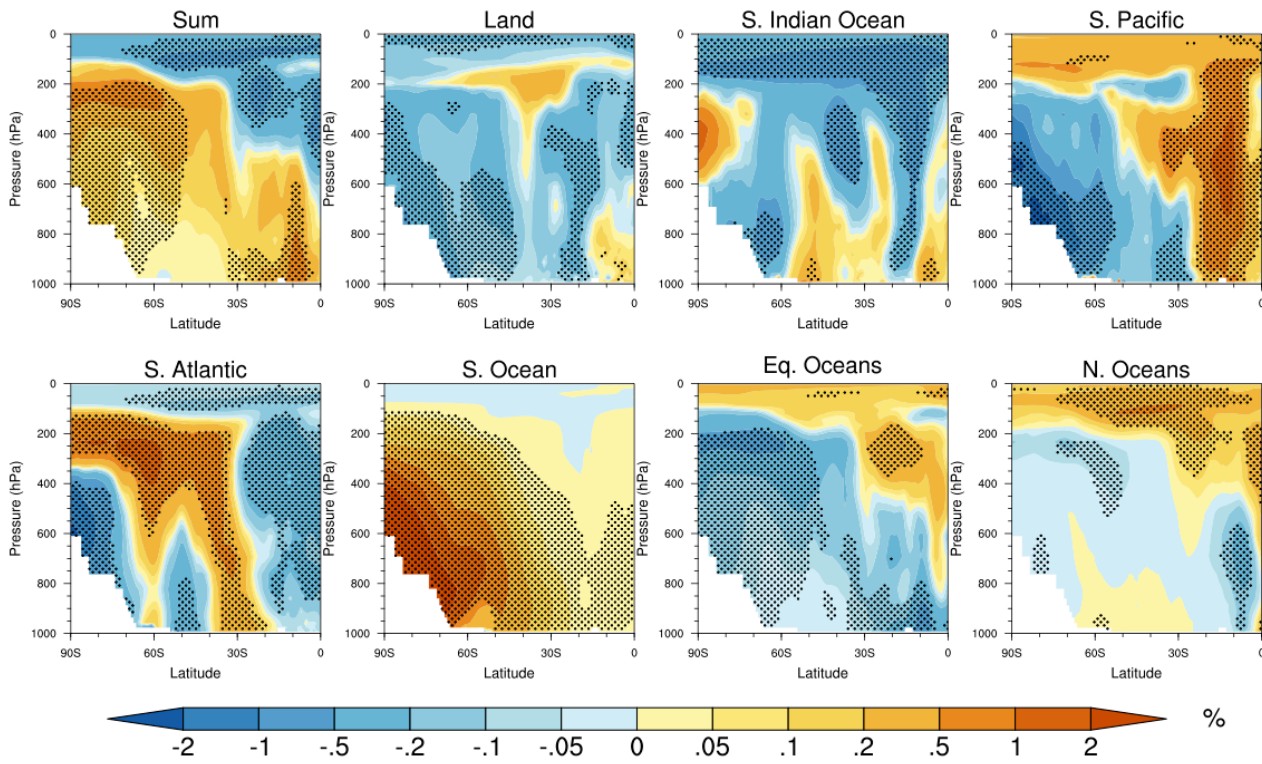

Figure 9: Vertical distribution of differences in fractional contribution to annual and zonal mean water vapor mixing ratio between the "low" and "high" SIC cases. Note that the contour intervals are non-uniform. Stippling indicates that the differences are statistically significant at the 90% confidence level based on Student's *t*-test.