# Peer review of "Influence of Sea Ice Anomalies on Antarctic Precipitation Using Source Attribution in the Community Earth System Model"

_The Cryosphere, 2019_

## Referee Comment (RC1) · Anonymous Referee #1 · 31 Jul 2019

In this article, Wang and co-authors use the atmospheric global climate model CAM5 to better understand the moisture origins and pathways toward Antarctica. This is a relevant scientific question for better interpreting ice cores and for better understanding the water cycle in the high Southern latitudes.

The novelty of this study is to use an explicit water source tagging capability in CAM5 to derive moisture sources of Antarctic precipitation. This cannot be done accurately with back-trajectories tools for long-ranged moisture transport.

I think this article is of interest, but conclusions must be deepened and I found substantial issues in the methods. Therefore, I recommend this article to be published in The Cryosphere after addressing the following issues.

**Major issues**

**1. Objectives of the paper are not clearly stated**

The introduction mixes issues related to the recent past (pre-industrial and historical=reanalyses periods, e.g. P3L10-13 and P3L24-P4L3) with issues related to future changes under RCP scenarios (e.g. P3 L13-17 and P4L3-7). However the article only deals with natural variability under pre-industrial conditions: the atmospheric global climat model CAM5 is ran with SIC and SST boundary condition taken from the pre-industrial control simulation of the CESM large ensemble (P5L21-30). And at the end, extreme « high » and « low » sea ice concentration (SIC) chosen from the CESM ensemble are very similar in winter (Fig. 1 JJA) and are divergent in the other seasons. This a major difference with future changes expected at the end of the 21st century, that will be driven by change in winter SIC (e.g. Agosta et al., 2015 as cited by the authors P4L5).

Consequently, the simulations performed by the authors can not address the impact of future change in SIC on the moisture pathway toward Antarctica, as driving mechanisms are expected to be different between pre-industrial and end-of-21C. For pre-industrial and historical periods, SMB changes and moisture pathways are dominated by strong natural variability, as stated P3 L13-17, whereas for large increase in temperature and decrease in winter SIC, as expected in future projections, SMB changes and moisture pathways are expected to be driven by the increase in moisture content, exponentially related to the increase in temperature (e.g. Krinner et al., 2014, Frieler et al, 2015).

I suggest to :

**1.i)** Explore the impact of future changes in SST/SIC on the moisture pathways, by performing new simulation using well chosen boundary condition among the CMIP5/6 datasets, and separate correctly issues related to internal variability vs. future changes in the introduction and the results.

**1.ii)** If these new simulations are not feasible, only focus on internal variability and remove all reference to future changes in the introduction. Better highlight the importance of better understanding water pathway toward Antarctica under natural variability, in particular for interpreting ice cores.

**1.iii)** Better exploit your current simulations to disentangle circulation changes vs. moisture input from SIC/SST changes (see Major issues 2 and 3 bellow).

Adapt the abstract consequently.

**2. The baseline simulation might not be valid and is not evaluated**

P5 L26-28 « A baseline simulation uses the mean SIC/SST distributions»

The baseline simulation is a simulation with CAM5 forced with mean SST and mean SIC from the CESM large ensemble. This is a major issue as SIC usually shows a sharp transition between SIC=1 and SIC=0. Averaging SIC across years/members lead to a SIC~0.5 over most of the Southern Ocean in all season except winter (Fig. 1). Combining mean SST and mean SIC might also lead to unrealistic boundary conditions, e.g. SST>-1.7 while SIC > 0.

As understanding water pathways toward Antarctic is one of the major goal of this study, that I think should be deepen, it is of importance that « baseline simulation » is proved to be relevant for analyzing current climate.

I suggest :

**2.i)**   To use either observed SIC and SST, or to use a median simulation for keeping realistic SIC and consistency between SIC and SST.

**2.ii)**   To show the differences in large scale circulation and SIC/SST between your 10-year simulations and reanalyses (1979-201X), to be able to analyze which of your conclusions may remain valid for the historical period, and which may not.

**3. Conclusions need to be deepened**

The goal of your study should be to disentangle the role of local moisture sources related to SIC/SST and the role of circulation on the moisture transport toward Antarctica. Currently this central point is not addressed by your study, as stated in the final sentences of the article.

**3.i)**   All analyses related to circulation (sea level pressure), precipitation amounts, precipitable water and moisture fluxes can be analysed with regard to the CESM ensemble from which SIC/SST have been extracted (e.g. P6L30-P7L25). From the CESM ensemble you can derive the decadal variability for each of these variables (e.g. standard deviation of 10-year-mean of each of these variable, by season). This will help quantifying the significance of precipitation changes between your three simulations with regard to decadal variability.

**P7 L5-7** « The sensible heat flux and evaporation over the northern latitudes of the SO also show large differences between the two cases, likely due to meteorological responses (e.g., changes in wind, temperature, and humidity) to the SIC/SST differences. »
You should remove « likely » and show here the difference in SLP and wind speed (your Fig. 10 should be Fig. 4). This should be the central point of your results and discussion.

**P7 L9-12** « The coastal area that has less precipitation in the low SIC case, mostly occurring in austral winter (JJA) when SIC near coastal regions is almost the same in the two cases (Fig. 1), is characterized by anomalous meridional moisture divergence (figure not shown), echoing the finding of Fyke et al. (2017). »
You can show it in your current Fig 10 (future Fig 4?).

**3.ii)**   You can exploit the fact that SIC changes between « low » and « high » SIC simulations are small in JJA and large in DJF to evaluate the impact of SIC change on circulation change. Indeed circulation changes between the 3 simulations might be due to changes in SIC but also due to internal(multidecadal) variability. This can be seen in Fig. 10 where circulation change

in DJF is of appear significant but of a lower magnitude than in JJA. If you do new simulations with decreased winter sea ice, it should have an impact of circulation too.

**3.iii)** To disentangle the effect of moisture source due to changed SST vs. circulation change on Antarctic precipitation, you can focus on the contribution of tagged regions that see large change in SIC, such as Weddell Sea, Amundsen Sea, Ross Sea, etc., in DJF, vs. regions that see no SIC change. The amount of Antarctic precipitation that is changed because of changed moisture uptake might be quantified from the contribution of the region of changed SIC vs. contribution of regions with unchanged SIC. Maps as in Fig. 5 but showing the difference between « high » and « low » sic, in JJA and DJF, would be useful for this interpretation.

**4. References**

**P3 L10-13** « However, the exact coupled-climate mechanisms driving this increase have not been well elucidated. In particular, the role of sea surface temperature (SST) changes, atmospheric moisture sources/transport/carrying capacity, sea ice loss, and atmospheric dynamical changes on Antarctic snowfall changes has not been clearly disaggregated. »

The increase in SMB expected at the end of the century is well understood: increase in moisture content is exponentially related to increase in temperature (~ Clausius Clapeyron). The relative contribution of thermodynamics (i.e. increase in temperature) vs. dynamics (i.e. circulation change) has already been addressed, e.g. in Krinner et al. (2014, doi: 10.1175/JCLI-D-13-00367.1), advocating for a major influence of thermodynamic changes in the future because of the expected large change in temperature.

**P3 L22-23** « However, the origin of moisture (i.e., evaporation source) and the impact of sea ice anomalies in the Southern Ocean on moisture source availability remain unclear. »

Kittel et al. (2018, doi:10.5194/tc-12-3827-2018) analyzed the impact of sea ice anomalies on the Antarctic surface mass balance, with circulation nudged toward a reanalysis, and they showed that only large anomalies of sea ice directly affect the Antarctic SMB.

**Minor comments**

*Abstract*

a climate model ↦ the global ocean-atmosphere coupled model XXX

SST: not introduced

Southern Ocean (SO): remove this acronyme from the abstract for readability

S. ↦ replace by South or Southern in the abstract

/year ↦ year⁻¹ (everywhere in the article)

"low" SIC case than in the "high" SIC case: rephrase with more explicit sentences

« so the contribution of nearby sources also depends on regional coastal topography »
I don't understand, why does the contribution of nearby sources depends on topography?

« The impact of sea ice anomalies on regional Antarctic precipitation also depends on atmospheric circulation changes that result from the prescribed composite SIC/SST perturbations. In particular, regional wind anomalies along with surface evaporation changes determine regional shifts in the zonal and meridional moisture fluxes that can explain some of the resultant precipitation changes. » This last sentence is very general. Can you write a sentence more specific about the novel knowledge brought by your study?

**Introduction**

P3 L3 « by supplying the vast majority of the positive mass component »
Is there other positive mass component?

P3 L4 « Lenaerts et al., 2012 »: an observation-based article would be better

P3 L5 Remove « this » and « profound »

P3 L7-8 « Frieler et al., 2015; Grieger et al., 2016; Lenaerts et al., 2016; Zwally et al., 2015; Medley and Thomas, 2019 »
Sort the list

P3 L8-10 « This SMB increase has the potential to offset a significant portion of the overall AIS mass loss due to ocean-driven mass loss (e.g., Winkelmann et al., 2012). »
This paper does not say this at all. Change the reference.

P3 L24-25 « Oceanic areas close to the Antarctic coast are ice-covered most of the year » : it is not true in austral summer (e.g. https://nsidc.org/data/seaice_index)

P4 L1 « natural or internal »
What do you mean?

**Methodology**

P5 L4-7 « The atmospheric component, called the Community Atmosphere Model version 5 (CAM5), can also be run with prescribed sea surface temperature (SST) and sea ice concentrations (SICs) coupled with an interactive land component (CLM4, Oleson et al., 2010), which includes the evolution of ice and snow over land. »
In this sentence, it is not clear that it is indeed the model setting you used. Please rephrase: *« In this study, we ran the atmospheric component of CESM, called the Community Atmosphere Model version 5 (CAM5), with prescribed sea surface temperature (SST) and sea ice concentrations (SICs) coupled, and with an interactive land component … »*

P5 L24-26 « Three SIC (and corresponding SST) composites are constructed from the pre-industrial control simulation of the CESM large ensemble (Kay et al., 2015), which gives a continuous time series of over 1000 years to perform our composite analysis of SIC and SST. »
Give more details on the CESM large ensemble, and at least the number of members.

P5 L26-28 « A baseline simulation uses the mean SIC/SST distributions and two sensitivity simulations use the 10% lowest and highest annual average total Southern Hemisphere SIC, respectively, coupled to the corresponding anomalies in global SSTs. »

Does it mean accross all members and all years, i.e. N members x 1000 years?
For mean SIC/SST, do you average it day-by-day to preserve the seasonal cycle? (as I guess from Fig. 1)

P5 L28-30 « All other forcing conditions (e.g., solar, greenhouse gases, anthropogenic aerosols) are identical across simulations. »

Set to which values ? pre-industrial ?

P6 L9 « Antarctic/SO »

SO defined latter in the text.

P6 L10 « CESM LENS »

Not defined, what is LENS?

P6 L15 « CESM »

CAM5? To clarify that you use CAM5 only and not the ocean-atmosphere coupled version CESM?

P6 L15 « Southern Ocean (SO) »

To be defined sooner. And be consistent all over the article, you often use « S. Ocean »

**Results and Discussions**

P7 L19 « 150 Gt/year »

Specify the mean precipitation in Gt year-1, and the area of your ice sheet mask (~14 M km2 ?)

P7 L26 « contributed »
contribution?

P9 L6-9 « As shown in previous studies (e.g., Wang et al., 2014; Singh et al., 2017), as well as indicated in the previous section (Fig. 5), the horizontal transport pathways of atmospheric constituents such as vapor and aerosol particles from individual source regions to a receptor are largely determined by large-scale atmospheric circulations. »

This is widely known indeed. Simplify the sentence, without citations.

P9 L12-15 « In general, vapor originating from remote source regions at lower latitudes and northern hemisphere takes elevated pathways to Antarctica while vapor from the nearby tags in the SO moves southward within the lower troposphere, as noted in previous studies (e.g., Noone and Simmonds, 2002; Sodemann and Stohl, 2009). »

See also Kittel et al. 2018

P9 L16 « mean moist isentropes moist »

Typo

P10 L19-20 « The pattern of variations in meridional moisture flux is also correlated with precipitation differences (Fig. 3f). »

Correlated? Precipitation is a result of large scale circulation... SLP represents the large scale circulation

P10 L20-22 « As a result, decreases in precipitation in the "low" SIC case over the King Haakon VII Sea and Wilkes Land sector can be traced to a SIC-caused reduction in meridional flow and related moisture fluxes from the north (Fig.10a). »

SIC-caused because SIC is the imposed boundary condition in CAM5, but how can you be sure it is not internal variability? In JJA it is arguable, as changes in SIC is the largest in this season and result in larger changes in SLP. But this should be discussed. How do you disentangled rigorously internal variability of the model vs. impact of changed SIC?

P10 L22-24 « Although the experimental design in this study doesn't allow us to pinpoint a causal relationship among the three effects (i.e., lower SIC, reduced meridional moisture flux, and precipitation decrease) »

Precipitation decrease is a result of circulation change.

P10 L29-31 « Therefore, the impact of sea ice anomalies and corresponding SST changes on Antarctic precipitation stem both from their direct impact on moisture sources and from the circulation changes that accompany the different SIC and SST patterns »

Your aim is to disentangle SIC change from circulation change. Here SIC does not change much in DJF, so changes in circulation cannot be attributed to changes in SIC.

P11 L11-13 « Conversely, the strength and location of the ABSL can also be affected by the sea ice and temperature changes, as depicted in Fig. 10e. »

For JJA only, and significance must be quantified vs. internal variability.

P11 L18-20 « In this study, we use a coupled atmosphere-land version of the Community Earth System Model (CESM1-CAM5) with explicit water tagging capability to quantify the impact of sea surface temperature (SST) and sea ice concentration (SIC) changes on the moisture sources of Antarctic precipitation. »

You use CAM5 with water tagging, not the coupled model. This sentence is confusing.

P11 L23 « 1800 »

Typo? 1000?

P11 L25-26 « are used as prescribed boundary conditions for atmosphere- only simulations. »

Add the length of simulations: 10 years

P11 L28-30 « Because of the prescribed changes in the SIC and SST, surface sensible heat fluxes and evaporation over the lower SIC areas in the Southern Ocean (SO) have a large increase in the "low" SIC case, compared to 30 the "high" SIC case. »

The relation with circulation change must be clarified here, significance of change must be quantified, depending of the season.

P12 L3-5 « The three remote source regions have a reduced absolute contribution to water vapor further inland in the "low" SIC case, which leads to a discernable reduction in their fractional contribution, especially, in the lower and mid troposphere. »

Because of change in circulation

P12 L7-8 « This is qualitatively consistent with the source attribution change in response to warming from CO2 doubling (Singh et al., 2017). »

Explain more.

P12 L9-12 « This difference is larger than the interannual variability of Antarctic precipitation (characterized by one standard deviation of annual mean precipitation) within the 10 years of the "mean" SIC case as well as over 1000 years of the CESM LENS experiment. »

And compared to decadal variability?

P12L32-P13L « The resultant changes in meridional moisture fluxes from the Southern Ocean to the Antarctic continent can intuitively explain some of the precipitation differences between the "low" and "high" SIC cases. »

Not quantified, very approximative

**Figures**

Figure 1

Don't use a divergent colorbar for sea ice concentration. Use a sequential colorbar instead.

Figure 2

Don't use a divergent colorbar for evaporation/sublimation, or around 0.

Display evaporation in kg m$^{-2}$ year$^{-1}$ (= mm year$^{-1}$)

Figure 3

Use symmetric colorscales around 0 (for Fsh, E, and P)

Change unit for E and P: kg m$^{-2}$ year$^{-1}$

Figure 4

Add a) and b) and change the main text accordingly

For precipitation, give the value in Gt year-1 or in Gt month-1, as in the text.

Figure 5, 6 and 7

Don't use divergent colorbars.

Figure 10

Use symmetric colorscales around 0.

> Same remarks for supplement

---

## Referee Comment (RC2) · Harald Sodemann (Referee) · 5 Aug 2019

**Review of "Influence of Sea Ice Anomalies on Antarctic Precipitation Using Source Attribution" by Wang et al., submitted to The Cryosphere.**

The authors present a sensitivity study of the global water cycle, testing the response of precipitation origin over Antarctica to combined sea ice cover and sea surface temperature changes, using a climate model capable of water vapour tagging. The results are interesting and novel, and fit well into the scope of TC. However, some aspects need further clarification, concerning the more general implications of the findings and the relation to previous results. In addition, the presentation quality of some figures can be improved, as detailed below.

[Figure]

**Major comments**

1. The findings should be placed more clearly into the wider scientific context to make their significance more obvious. This concerns the abstract, introduction and conclusions.

2. The relation to previous studies should be more clearly specified. The present study is based on a long control run, whereas previous studies mostly used reanalysis data. What are the differences? How valid is the control run for the present-day simulation? And, more specifically, how do the uppermost and lowermost percentiles used here compare to observed natural variability?

3. The tagging setup should to be modified, or justified more clearly, and be documented more comprehensively. The Antarctic land mass is currently part of a general land tracer in the tagging setup. It would be interesting to specify an Antarctic-land only tracer, to allow for distinguishing local from remote contributions. In addition, the Southern Ocean tag appears specified in a confusing way, with a narrow stripe going around the globe, and some boxes for the Weddell Sea and others placed inside. It should be stated more clearly why this specific setup has been chosen. Furthermore, a table or other form of description of the exact box coordinates are needed in order to compare and reproduce results.

4. The findings in some figures should be condensed further to facilitate grasping the main findings, as detailed in the specific comments

**Specific comments**

*Abstract*

Pg. 2, L. 1: highlight the relevance to other research, now it is just described as a sensitivity study in the first sentence.

Pg. 2, L. 6: "but": consider splitting into two sentences

Pg. 2, L. 11: A percent number could be more informative here

*Introduction*

Pg. 3, L. 1: "SMB ... plays a role... by supplying" does not make sense

Pg. 3, L. 10-13: Not sure the exact mechanism is assessed quantitatively in this study. May need rephrasing.

Pg. 3, L. 17: "Because of..." ultimate meaning/logic of this sentence not clear. "Relies" for what purpose? Maybe it helps to rephrase in terms of *mean* temperature?

Pg. 3, L. 20: The relevance of knowing the moisture source could be highlighted here.

Pg. 3, L. 24 onward: Some link to the state of the Antarctic hydrologic system from observation or reanalysis data would be useful here (e.g., Papritz et al., 2014).

Pg. 4, L. 1: Clarify the distinction between internal and natural climate variability

Pg. 4, L. 2: "such responses": unclear what exactly is referred to

Pg. 4, L. 10: "and/or" - rephrase

Pg. 4, L. 17: "unacceptably" - I think this depends on the approach and purpose. One frequently used countermeasure is to consider the problem stochastically, by calculating many trajectories, as is done in Sodemann and Stohl (2009). Interpreting single trajectories beyond 10 days in contrast may be meaningless. On the other hand, tracer studies suffer from numerical diffusion and the uncertainty of parameterisation processes, and do not provide the spatial detail of source contributions available from backward trajectories. A more balanced discussion would be justified here.

Pg. 4, L. 20 onward: The relevance for ice core studies could be included as an additional motivation, see for example Winkler et al., 2012, Wang et al., 2013, Masson-Delmotte et al., 2011, Buizert et al., 2018 and references therein.

[Figure]

*Methods*

Pg. 4, L. 22: "such studies" - which ones specifically?

Pg. 5, L. 3: Are all 5 references needed as reference to the method, or rather example applications? Please clarify.

Pg. 6, L. 3: "assuming" - has it been checked that the simulation stabilises after 1 year?

Pg. 6, L. 13: It would be helpful to more convincingly illustrate and quantify this aspect.

Pg. 6, L. 23: tags -> tag

Pg. 6, L. 25: remove "well"

Pg. 6, L. 26: "differencing": rephrase, e.g. distinguishing?

Pg. 6, L. 15 onward: SST mean and anomalies should be shown/discussed and placed in relation to observations/reanalysis.

Pg. 6, L. 15: see major comment 3.

*Results*

Pg. 7, L. 11: "echoing the finding" - what aspect specifically?

Pg. 7, L. 20: This number could be useful to include in the Abstract

Pg. 7, L. 22 onward: would be helpful to put these numbers into the context of observation/reanalysis data

Pg. 7, L. 27: what is meant by "more significant"?

Pg. 8, L. 1: It would be interesting to know what land region contributes, in particular anything from Antarctica.

Pg. 8, L, 2 onward: it would be helpful to collect these results in a table.

Pg. 8, L. 8: The discussion could be more comprehensive. My take is that the overall results are quite similar, and specific numbers depend on how the ocean sectors have been defined here and in Sodemann and Stohl (2009). That itself is a useful finding that should be stated clearly. Part of the differences may then be due to the fact that Sodemann and Stohl (2009) based their results on ECMWF analyses for a specific period, while you consider a control run of a climate model. Comparison to reanalysis data may therefore be helpful to better explain the differences found here. Furthermore, a table or other form of description of the exact box coordinates are needed in order to compare (and reproduce) results.

Pg. 8, L. 30: Fig S4 seems to contain useful results but the information needs to be condensed more (see below).

Pg. 9, L. 9: See, for example, Stohl and Sodemann (2010), which also clearly illustrates the thermodynamic transport barrier to low-level airmasses from the Southern Ocean (their Fig. 3).

Pg. 9, L. 9: "at the source" - or underways!

Pg. 9, L. 17: The presentation of the results can be improved, see comments on figures below.

Pg. 9, L. 25 onward: The discussion here is quite vague, and could be made more specific

Pg. 10, L. 4 onward: What is the region for the apparently substantial changes in the NH? Maybe better to only show the SH.

Pg. 10, L. 30: It appears you imply a direct and indirect impact from the SIC and SST anomalies - if this is a main result, this should be introduced more prominently already in the Introduction. Is it really possible to separate both aspects, as dynamic changes can also affect evaporation of the source regions?

Pg. 11, L. 15: The takeaway from this paragraph is not fully clear.

*Summary*

Pg. 12, L. 15: It could be helpful to state whether or not this confirms earlier findings (to my understanding it does)

Pg. 12, L. 34: "intuitively" - may not be applicable to all readers

Pg. 13, L. 1-5: Would be useful to highlight the wider implications of this study in the end.

*Figures*

Fig. 1: In Antarctica, seasons are more commonly defined as JFM and ASO, in line with the sea ice seasonaliry. Maybe it would probably be sufficient to show these seasons only along with the annual mean.

Fig. 2: The E panel does not appear to be relevant here, but could be part of a "validation" figure where you compare SH plots of annual mean P and E from the model simulation with reanalysis/observation data.

Fig. 4: This figure could be made more easily readable by either showing bars for 3-month periods, or by removing the white space in between the individual monthly bars. Similarly, Fig. S4 could be made into a much simpler figure that only compares the annual mean or summer/winter differences for the 3 regions.

Fig. 5: Panel a is method validation and could be removed in this context, the big red spot draws a lot of attention, and prevents a more useful color scale to be used. For the purpose of the paper, it seems it would be more useful to highlight the contributions to precipitation in Antarctica only, by masking the tracer contribution over the Oceanic regions, and zooming in on Antarctica.

Fig. 6: Instead of a zonal mean covering all latitudes, it would be more useful to highlight the Southern Hemisphere only. In fact, Fig. 6 may be dropped altogether, and be replaced by Fig. S5.

Fig. 7: Similarly, the figure would be improved by showing the SH part only.

Fig. 8: May be dropped in favour of Fig. 9

Fig. 9: Show SH section only (or discuss the NH changes which are in general larger or as large as the NH changes)

Fig. 10: Busy figure with 15 panels and the SLP difference contours. Consider removing panel c and d, and e, or keeping e and removing the contours from the other panels.

*References*

Buizert, Christo; Sigl, Michael; Severi, Mirko; Markle, Bradley; Wettstein, Justin; McConnell, J; Pedro, Joel; Sodemann, H.; Goto-Azuma, Kumiko; Kawamura, Kenji; Fujita, Shuji; Motoyama, Hideaki; Hirabayashi, Motohiro; Uemura, Ryu; Stenni, Barbara; Parrenin, Frederic; He, Feng; Fudge, T.J.; Steig, Eric J., 2018: Abrupt ice-age shifts in southern westerly winds and Antarctic climate forced from the north, Nature 563: 681-685.

Masson-Delmotte, V., Buiron, D., Ekaykin, A., Frezzotti, M., Gallée, H., Jouzel, J., Krinner, G., Landais, A., Motoyama, A., Oerter, H., Pol, K., Pollard, D., Ritz, C., Schlosser, E., Sime, L. C., Sodemann, H., Stenni, B., Uemura R., and Vimeux, F., 2011: A comparison of the present and last interglacial periods in six Antarctic ice cores Clim. Past, 7, 397-423.

Papritz, L., Pfahl, S., Rudeva, I., Simmonds, I., Sodemann, H., and Wernli, H., 2014: The role of extratropical cyclones and fronts for Southern Ocean freshwater fluxes, J. Climate 27: 6205–6224, doi:10.1175/JCLI-D-13-00409.1.

Stohl, A., and Sodemann, H., 2010: Characteristics of atmospheric transport into the Antarctic troposphere, J. Geophys. Res., 115, D02305, doi:10.1029/2009JD012536.

Wang, Y., Sodemann, H., Hou, S., Masson-Delmotte, V. Jouzel, J. and Pang, H., 2013:

Snow accumulation and its moisture origin over Dome Argus, Antarctica. Clim. Dyn., 40:731-742, doi: 10.1007/s00382-012-1398-9.

Winkler, R., Landais, A., Sodemann, H., Dümbgen, L., Priéa, F., Masson-Delmotte, V., Stenni, B., and Jouzel, J., 2012: Deglaciation records of 17O-excess in East Antarctica: reliable reconstruction of oceanic relative humidity from coastal sites. Clim. Past 8, 1-16.

---

## Referee Comment (RC3) · Anonymous Referee #3 · 12 Aug 2019

General Comments:

As far as it goes, this exploration of the impact of sea ice extent anomalies on Antarctic precipitation within the climate model context is competently done and the findings are interesting and valuable. Unfortunately, there are many things left dangling that require some effort to rectify before publication.

1. This is a model study and the title should reflect this. 2. There is very little effort made to relate the results to the real world, rather the manuscript seems to assume that the results must be realistic. What constraints can you apply throughout the manuscript to the results to verify their credibility? In the background, stable water isotope studies are relevant, so it would be nice to see explicit discussion of relevant results. 3. Explain why you used a pre-industrial control as the basis for your atmospheric sensitivity

studies. What difference does this make to today? Do you think that fixed SST and sea ice distort your results in contrast to having an interactive ocean? 4. There are some unexplained results for low versus high sea ice. What is the reason large PW increase north of 55S between 90E and 120E (Fig. 3c)? Why does the surface sensible heat flux decrease north of 55s (Fig. 3d)? Why does the latent heat flux decrease north of 55S between 90E and 170E (Fig. 3e)? This impacts the precipitation (Fig. 3f). You could discuss/explain these results after presenting Fig. 10. 5. You use unusual units for P, g/m*m/h, in contrast to the frequent mm/d or mm/yr. The latter units allow the reader to evaluate the magnitude of the simulated changes.

Smaller Comments.

6. Page 3, line 17: Palerme et al. (2016) as per reference list? 7. Page 4, line: Difference between natural and internal climate variability? 8. Page 5, line 2: Need Hurrell et al. (2013) reference. 9. Page 6, line 8: "further" than what? 10. Page 6, line 10: What is "CESM LENS"? 11. Page 7, line 11: Must be "anomalous meridional moisture transport divergence" to fit with the atmospheric water balance equation. 12. Page 7, line 18: Fall and spring are when the low-pressure trough around Antarctica is closest to the continent, known as the semi-annual oscillation. 13. Page 8, line 1: "Evaporation/sublimation over land". This is a quite surprising result. Do you mean primarily over Antarctica in summer? 14. Page 8, line 24: Why do you think that the remote sources mostly lead to precipitation decreases? 15. Figs. 3, 8-10, S2, S3: Statistical significance should be tested for these figures. 16. Fig. 10f: Please use the more physically meaningful hPa rather than Pa for SLP differences.

---

## Author Comment (AC1) · 1 Oct 2019

We appreciate the insightful review and constructive comments from all three referees. The paper has been improved after addressing all the comments and concerns. Below we outline the point-by-point responses and changes made to the manuscript. Direct changes to the paper are italicized and all figures referenced are either within this document, in the original manuscript, or for the updated version of the manuscript. The responses to Referee #2 begin on page 15 and the responses to Referee #3 begin on page 24.

**Anonymous Referee #1**

In this article, Wang and co-authors use the atmospheric global climate model CAM5 to better understand the moisture origins and pathways toward Antarctica. This is a relevant scientific question for better interpreting ice cores and for better understanding the water cycle in the high Southern latitudes. The novelty of this study is to use an explicit water source tagging capability in CAM5 to derive moisture sources of Antarctic precipitation. This cannot be done accurately with back-trajectories tools for long-ranged moisture transport. I think this article is of interest, but conclusions must be deepened and I found substantial issues in the methods. Therefore, I recommend this article to be published in The Cryosphere after addressing the following issues.

Please see our responses to the specific comments below.

**Major issues**

1. **Objectives of the paper are not clearly stated**

The introduction mixes issues related to the recent past (pre-industrial and historical=reanalyses periods, e.g. P3L10-13 and P3L24-P4L3) with issues related to future changes under RCP scenarios (e.g. P3 L13-17 and P4L3-7). However the article only deals with natural variability under pre- industrial conditions: the atmospheric global climat model CAM5 is ran with SIC and SST boundary condition taken from the pre-industrial control simulation of the CESM large ensemble (P5L21-30). And at the end, extreme « high » and « low » sea ice concentration (SIC) chosen from the CESM ensemble are very similar in winter (Fig. 1 JJA) and are divergent in the other seasons.

This a major difference with future changes expected at the end of the 21st century, that will be driven by change in winter SIC (e.g. Agosta et al., 2015 as cited by the authors P4L5).

Consequently, the simulations performed by the authors can not address the impact of future change in SIC on the moisture pathway toward Antarctica, as driving mechanisms are expected to be different between pre-industrial and end-of-21C. For pre-industrial and historical periods, SMB changes and moisture pathways are dominated by strong natural variability, as stated P3 L13-17, whereas for large increase in temperature and decrease in winter SIC, as expected in future projections, SMB changes and moisture pathways are expected to be driven by the

increase in moisture content, exponentially related to the increase in temperature (e.g. Krinner et al., 2014, Frieler et al, 2015).

I suggest to :

**1.i)** Explore the impact of future changes in SST/SIC on the moisture pathways, by performing new simulation using well chosen boundary condition among the CMIP5/6 datasets, and separate correctly issues related to internal variability vs. future changes in the introduction and the results.

**1.ii)** If these new simulations are not feasible, only focus on internal variability and remove all reference to future changes in the introduction. Better highlight the importance of better understanding water pathway toward Antarctica under natural variability, in particular for interpreting ice cores.

**1.iii)** Better exploit your current simulations to disentangle circulation changes vs. moisture input from SIC/SST changes (see Major issues 2 and 3 bellow).

Adapt the abstract consequently.

This is a great point. By design our current simulations were conducted to separate the impact of Southern Ocean SIC/SST anomalies from the effect of anthropogenic warming (e.g., associated with GHGs) on source–receptor relationships for moisture and precipitation over Antarctica. Thus we took the SIC and SST anomalies from the pre-industrial control simulation of CESM Large Ensemble simulations. In a previous study, Singh et al. (2017) examined the response of Antarctic hydrological cycle to warming from $CO_2$ doubling. As we discussed in the manuscript (P12L7), some of the qualitative results in terms of the water source attribution are consistent with their findings (under the GHGs warming scenario). Similarly, we referred to some previous studies (mostly in the introduction) that can provide a broader context for the motivation of the present study to focus on the isolated impact of SIC/SST anomalies associated with internal climate variability. Nonetheless, we have followed the referee's suggestions (ii, iii) to clarify the objective of the present study and revise the manuscript accordingly. In particular, we have made a note in Section 2.2 (Experimental Design) as follows.

*"Although the magnitude and location of prescribed SIC anomalies are comparable to the observed SIC changes during the recent decades (Hobbs et al., 2016), the prescribed seasonal SIC anomalies associated with internal variabilities under the CESM LENS pre-industrial conditions are likely to be different from future changes. Here the widespread anomalies occur in austral summer (DJF), while sea ice reductions by the end of 21st century or in response to $CO_2$-doubling are expected to be dominated by winter (JJA) changes. Therefore, the simulations designed here are to examine Antarctic precipitation changes and moisture transport pathways dominated by natural variabilities, as opposed to the projected future changes driven by the increase in atmospheric moisture content related to temperature increase (e.g. Krinner et al., 2014, Frieler et al, 2015)."*

**2. The baseline simulation might not be valid and is not evaluated**

P5 L26-28 « A baseline simulation uses the mean SIC/SST distributions»

The baseline simulation is a simulation with CAM5 forced with mean SST and mean SIC from the CESM large ensemble. This is a major issue as SIC usually shows a sharp transition between SIC=1 and SIC=0. Averaging SIC across years/members lead to a SIC~0.5 over most of the Southern Ocean in all season except winter (Fig. 1). Combining mean SST and mean SIC might also lead to unrealistic boundary conditions, e.g. SST>-1.7 while SIC > 0.

As understanding water pathways toward Antarctic is one of the major goal of this study, that I think should be deepen, it is of importance that « baseline simulation » is proved to be relevant for analyzing current climate.

I suggest:

**2.i)** To use either observed SIC and SST, or to use a median simulation for keeping realistic SIC and consistency between SIC and SST.

**2.ii)** To show the differences in large scale circulation and SIC/SST between your 10-year simulations and reanalyses (1979-201X), to be able to analyze which of your conclusions may remain valid for the historical period, and which may not.

We would respectfully argue that climatological SST and SIC for individual months being used in atmospheric GCMs are normally averages over many years as well. We can imagine that the sharp transition between SIC=1 and SIC=0 can happen in the real world, but in coarse-grid models we don't see such a sharp transition near sea ice edges, where sea ice concentrations, if being prescribed, are often interpreted from two monthly mean datasets at each model time step.

Nonetheless, we have taken the referee's suggestion to compare our baseline simulation to the fifth generation ECMWF reanalysis (ERA5, 1979-2018). All the spatial distribution fields showed in the original Figs. 3 and 10 are analyzed and included in the Supplement (new Figs. S1-S4). Annual mean SIC, surface temperature and sea level pressure are shown here (see Fig. R1 below). SIC in the baseline simulation is apparently higher than the ERA5 reanalysis, especially in the Weddell, Amundsen and Ross seas where the SIC internal variability (difference between the "low" and "high" SIC cases) is also large. Certainly, SIC in the ERA5 reanalysis reflects anthropogenic forcing already. Surface temperature differences are consistent with SIC differences. We don't see any unrealistic boundary conditions that the referee concerned about. The SLP pattern in the baseline simulation resembles the one in ERA5, although the magnitude of SLP gradient between Antarctic and the Southern Ocean is weaker in the baseline simulation. The comparison suggests that our pre-industrial simulations of the model sensitivity to SIC anomalies are still relevant for the recent historical conditions. We have now included some discussions on this in the manuscript (Section 2.2).

[Figure]

Fig. R1: Annual mean (a) sea ice concentrations, (b) surface temperature, and (c) sea level pressure based on the baseline simulation (top) and the ERA5 1979-2018 reanalysis (bottom).

**3.     Conclusions need to be deepened**

The goal of your study should be to disentangle the role of local moisture sources related to SIC/ SST and the role of circulation on the moisture transport toward Antarctica. Currently this central point is not addressed by your study, as stated in the final sentences of the article.

**3.i)**     All analyses related to circulation (sea level pressure), precipitation amounts, precipitable water and moisture fluxes can be analysed with regard to the CESM ensemble from which SIC/SST have been extracted (e.g. P6L30-P7L25). From the CESM ensemble you can derive the decadal variability for each of these variables (e.g. standard deviation of 10-year-mean of each of these variable, by season). This will help quantifying the significance of precipitation changes between your three simulations with regard to decadal variability.

Following the referee's suggestion, we have now calculated the decadal variability (based on the 1100-year CESM-LENS control simulation) of all fields in the original Figures 3 and 10 by

annual mean and seasonal (DJF and JJA) mean. The new analyses for comparison between the model results and decade variability are now included in in the Supplement (as new Figs. S6-S11). The figures are referred to in the manuscript to indicate the significance of differences in the relevant model variables between the "low" and "high" SIC cases.

**P7 L5-7** « The sensible heat flux and evaporation over the northern latitudes of the SO also show large differences between the two cases, likely due to meteorological responses (e.g., changes in wind, temperature, and humidity) to the SIC/SST differences. »

You should remove « likely » and show here the difference in SLP and wind speed (your Fig. 10 should be Fig. 4). This should be the central point of your results and discussion.

Changed as suggested. The original Fig. 10 has been revised and now becomes the new Fig. 4. The original Section 3.4 that describes the figure of circulation and moisture transport is moved up accordingly.

**P7 L9-12** « The coastal area that has less precipitation in the low SIC case, mostly occurring in austral winter (JJA) when SIC near coastal regions is almost the same in the two cases (Fig. 1), is characterized by anomalous meridional moisture divergence (figure not shown), echoing the finding of Fyke et al. (2017). »

You can show it in your current Fig 10 (future Fig 4?).

This has been included in the new Fig. 4.

**3.ii)** You can exploit the fact that SIC changes between « low » and « high » SIC simulations are small in JJA and large in DJF to evaluate the impact of SIC change on circulation change.

Indeed circulation changes between the 3 simulations might be due to changes in SIC but also due to internal(multidecadal) variability. This can be seen in Fig. 10 where circulation change in DJF is of appear significant but of a lower magnitude than in JJA. If you do new simulations with decreased winter sea ice, it should have an impact of circulation too.

We agree that the changes in circulation and moisture transport can be due to both SIC/SST differences and internal variability. As shown in the new Figs. S7 and S9, SIC differences in DJF is more widespread (e.g., large SIC changes near coastal areas) than in JJA, while the JJA SIC changes are concentrated at sea ice edges. The contrast in surface temperature and heat fluxes within the sea ice zone is much stronger in JJA than in DJF, so does the SLP over Southern Ocean and Antarctica. However, the decadal variability of these fields is also stronger in JJA than in DJF (See Figs. S8, S10 and S11). It indicates that decadal variability plays a more important role in JJA than in DJF, but it is still challenging to quantitatively compare the contributions by decadal variability vs. SIC/SST anomalies. We have included some discussions on this in the manuscript.

**3.iii)**     To disentangle the effect of moisture source due to changed SST vs. circulation change on Antarctic precipitation, you can focus on the contribution of tagged regions that see large change in SIC, such as Weddell Sea, Amundsen Sea, Ross Sea, etc., in DJF, vs. regions that see no SIC change. The amount of Antarctic precipitation that is changed because of changed moisture uptake might be quantified from the contribution of the region of changed SIC vs. contribution of regions with unchanged SIC. Maps as in Fig. 5 but showing the difference between « high » and « low » sic, in JJA and DJF, would be useful for this interpretation.

We have now made new JJA and DJF figures, showing the difference in contributions to Antarctic precipitation between "low" and "high" SIC cases, to compare different regions. They are included in the Supplement (Figs. S14, S15, S17 and S18). The challenge is that SIC reduction is ubiquitous in the Southern Ocean. There is no clean contrast in the source contribution between changed and unchanged SIC regions. However, the DJF and JJA figures more clearly demonstrate the important role of meridional flow in bringing the additional moisture from SIC reduction to Antarctica. For example, with the decrease in SIC, contributions from the Weddell Sea increase near the Antarctic Peninsula in both DJF and JJA under the favorite change in meridional moisture transport, while there is a strong seasonal contrast in contribution to precipitation near the Queen Maud Land because of the difference in meridional flow. Similar seasonal contrast is also seen in the contributions from Cosmonauts Sea and Amundsen Sea. We have added such discussions to the manuscript.

**4.     References**

**P3 L10-13** « However, the exact coupled-climate mechanisms driving this increase have not been well elucidated. In particular, the role of sea surface temperature (SST) changes, atmospheric moisture sources/transport/carrying capacity, sea ice loss, and atmospheric dynamical changes on Antarctic snowfall changes has not been clearly disaggregated. »

The increase in SMB expected at the end of the century is well understood: increase in moisture content is exponentially related to increase in temperature (~ Clausius Clapeyron). The relative contribution of thermodynamics (i.e. increase in temperature) vs. dynamics (i.e. circulation change) has already been addressed, e.g. in Krinner et al. (2014, doi: 10.1175/JCLI-D-13-00367.1), advocating for a major influence of thermodynamic changes in the future because of the expected large change in temperature.

Added the reference and revised the sentence accordingly.

**P3 L22-23** « However, the origin of moisture (i.e., evaporation source) and the impact of sea ice anomalies in the Southern Ocean on moisture source availability remain unclear. »

Kittel et al. (2018, doi:10.5194/tc-12-3827-2018) analyzed the impact of sea ice anomalies on the Antarctic surface mass balance, with circulation nudged toward a reanalysis, and they showed that only large anomalies of sea ice directly affect the Antarctic SMB.

Thanks for pointing out this reference. We have now incorporated the key finding of Kittel et al. (2018) in the introduction to provide a more appropriate context for our study. It is revised as follows:

*Kittel et al. (2018) conducted sensitivity experiments in a regional climate model, with atmospheric circulation nudged toward a reanalysis, to study the impact of SIC/SST perturbations on AIS SMB, and they found significant SMB anomalies for the largest combined SIC/SST perturbations. By analyzing long quasi-equilibrium global climate model simulations, Fyke et al. (2017) identified statistically significant relationships in Antarctic basin-scale precipitation patterns that are driven by variability in large-scale atmospheric moisture transport. However, the origin of moisture (i.e., evaporation source of remote and nearby ocean basins) and the impact of sea ice anomalies in the Southern Ocean (SO) associated with internal variability on Antarctic moisture source apportionment as well as their feedback on atmospheric circulation remain unclear.*

**Minor comments**

***Abstract***

a climate model ↦ the global ocean-atmosphere coupled model

Changed to "general circulation model" since SST and SIC are prescribed to the model.

XXX SST: not introduced

Now defined.

Southern Ocean (SO): remove this acronyme from the abstract for readability

Done.

S. ↦ replace by South or Southern in the abstract

Done.

/year ↦ year-1 (everywhere in the article)

Done.

"low" SIC case than in the "high" SIC case: rephrase with more explicit sentences

Rephrased to "Comparing two experiments prescribed with high and low pre-industrial SIC, respectively, the annual mean Antarctic precipitation is about 150 Gt year$^{-1}$ more in the lower SIC case than in the higher SIC case."

« so the contribution of nearby sources also depends on regional coastal topography »

I don't understand, why does the contribution of nearby sources depends on topography?

As indicated in the previous sentence and discussed more in the main text, the meridional and vertical transport of vapor is along moist isentropes ($\theta$e) that are largely shaped by local topography in Antarctica (see the original Fig. S5). More precisely, high elevation of coastal

mountains can block the transport moisture from nearby sources.

« The impact of sea ice anomalies on regional Antarctic precipitation also depends on atmospheric circulation changes that result from the prescribed composite SIC/SST perturbations. In particular, regional wind anomalies along with surface evaporation changes determine regional shifts in the zonal and meridional moisture fluxes that can explain some of the resultant precipitation changes. »

This last sentence is very general. Can you write a sentence more specific about the novel knowledge brought by your study?

Revised this sentence and added a closing statement: "This study highlights the importance of better understanding changes in water transport toward Antarctica under natural variability."

**Introduction**

P3 L3 « by supplying the vast majority of the positive mass component » Is there other positive mass component?

Removed "the vast majority of".

P3 L4 « Lenaerts et al., 2012 »: an observation-based article would be better

Added Shepherd et al. (2012).

P3 L5 Remove « this » and « profound »

Done.

P3 L7-8 « Frieler et al., 2015; Grieger et al., 2016; Lenaerts et al., 2016; Zwally et al., 2015; Medley and Thomas, 2019 »

Sort the list

Done.

P3 L8-10 « This SMB increase has the potential to offset a significant portion of the overall AIS mass loss due to ocean-driven mass loss (e.g., Winkelmann et al., 2012). »

This paper does not say this at all. Change the reference.

The sentence has been removed in response to the major comment #1.

P3 L24-25 « Oceanic areas close to the Antarctic coast are ice-covered most of the year » : it is not true in austral summer (e.g. https://nsidc.org/data/seaice_index)

The sentence has been revised to "… most of the year (except for austral summer)."

P4 L1 « natural or internal» What do you mean?

Both "natural" and "internal" refer to "unforced" climate variability. To avoid confusion, it has been changed to "internal climate variability".

**Methodology**

P5 L4-7 « The atmospheric component, called the Community Atmosphere Model version 5 (CAM5), can also be run with prescribed sea surface temperature (SST) and sea ice concentrations (SICs) coupled with an interactive land component (CLM4, Oleson et al., 2010), which includes the evolution of ice and snow over land. »

In this sentence, it is not clear that it is indeed the model setting you used. Please rephrase: *« In this study, we ran the atmospheric component of CESM, called the Community Atmosphere Model version 5 (CAM5), with prescribed sea surface temperature (SST) and sea ice concentrations (SICs) coupled, and with an interactive land component ... »*

Revised as suggested.

P5 L24-26 « Three SIC (and corresponding SST) composites are constructed from the pre-industrial control simulation of the CESM large ensemble (Kay et al., 2015), which gives a continuous time series of over 1000 years to perform our composite analysis of SIC and SST. »

Give more details on the CESM large ensemble, and at least the number of members.

The fully coupled pre-industrial control simulation was conducted for 1500 years with years 400-1500 released. It's considered as one member of the Large Ensemble. Transient simulations (1920-2100) have 30 members but they are not so relevant to the control simulation used in this study. Nonetheless, we have added more details as follows:

*"Three SIC (and corresponding SST) composites are constructed from the pre-industrial control simulation of the CESM Large Ensemble (hereafter CESM LENS; Kay et al., 2015), which was initialized with January mean present-day potential temperature and salinity from Polar Science Center Hydrographic Climatology dataset for ocean and a previous CESM 1850 control run for atmosphere, land and sea ice. It was run for 1500 years with years 400-1500 released, giving a continuous time series of over 1000 years to perform our composite analysis of SIC and SST."*

P5 L26-28 « A baseline simulation uses the mean SIC/SST distributions and two sensitivity simulations use the 10% lowest and highest annual average total Southern Hemisphere SIC, respectively, coupled to the corresponding anomalies in global SSTs. »

Does it mean accross all members and all years, i.e. N members x 1000 years?

For mean SIC/SST, do you average it day-by-day to preserve the seasonal cycle? (as I guess from Fig. 1)

Yes, it means across all years for the baseline simulation and about 100 years for the two sensitivity simulations. The mean SIC/SST was obtained from monthly means to preserve the seasonal cycle. We have now clarified this in the paper.

P5 L28-30 « All other forcing conditions (e.g., solar, greenhouse gases, anthropogenic aerosols) are identical across simulations. »

Set to which values? pre-industrial?

Pre-industrial conditions.

P6 L9 « Antarctic/SO » SO defined latter in the text.

Fixed it.

P6 L10 « CESM LENS »

Not defined, what is LENS?

It stands for Large Ensemble simulations. Now defined in Section 2.2.

P6 L15 « CESM »

CAM5? To clarify that you use CAM5 only and not the ocean-atmosphere coupled version CESM?

Yes, it has been changed to CAM5.

P6 L15 « Southern Ocean (SO) »

To be defined sooner. And be consistent all over the article, you often use « S. Ocean »

Done. Southern Ocean (SO) refers to the geographical region. S. Ocean is used for the tagged source region.

*Results and Discussions*

P7 L19 « 150 Gt/year »

Specify the mean precipitation in Gt year-1, and the area of your ice sheet mask (~14 M km2 ?)

The mean precipitation is 2500 Gt year-1, as noted later in the parentheses later in the same sentence. It was calculated for the entire Antarctic using the land mask from the same model.

P7 L26 « contributed » contribution?

It means S. Ocean contribution (i.e., the amount of precipitation contributed by the S. Ocean tag). It's now clarified.

P9 L6-9 « As shown in previous studies (e.g., Wang et al., 2014; Singh et al., 2017), as well as indicated in the previous section (Fig. 5), the horizontal transport pathways of atmospheric constituents such as vapor and aerosol particles from individual source regions to a receptor are largely determined by large-scale atmospheric circulations. »

This is widely known indeed. Simplify the sentence, without citations.

The sentence has been simplified to "As indicated in the previous section (Fig. 5), the horizontal transport pathways of atmospheric water from individual source regions to a receptor are largely determined by large-scale atmospheric circulations."

P9 L12-15 « In general, vapor originating from remote source regions at lower latitudes and northern hemisphere takes elevated pathways to Antarctica while vapor from the nearby tags in the SO moves southward within the lower troposphere, as noted in previous studies (e.g., Noone and Simmonds, 2002; Sodemann and Stohl, 2009). » See also Kittel et al. 2018

This reference has been added to the list.

P9 L16 « mean moist isentropes moist » Typo

Corrected.

P10 L19-20 « The pattern of variations in meridional moisture flux is also correlated with precipitation differences (Fig. 3f). »

Correlated? Precipitation is a result of large scale circulation... SLP represents the large scale circulation

We agree with the referee that there is a causal relationship between atmospheric circulation and large-scale precipitation, but not necessarily for meridional moisture flux and total precipitation over the broad area. Also, here we don't want to make the generic claim without providing concrete analysis. Later in the same paragraph we tried to pinpoint a causal relationship between lower SIC, reduced meridional moisture flux, and precipitation decrease for specific regions.

P10 L20-22 « As a result, decreases in precipitation in the "low" SIC case over the King Haakon VII Sea and Wilkes Land sector can be traced to a SIC-caused reduction in meridional flow and related moisture fluxes from the north (Fig.10a). »

SIC-caused because SIC is the imposed boundary condition in CAM5, but how can you be sure it is not internal variability? In JJA it is arguable, as changes in SIC is the largest in this season and result in larger changes in SLP. But this should be discussed. How do you disentangled rigorously internal variability of the model vs. impact of changed SIC?

Please also see our response to major comment #3. We agree that we cannot rule out the impact of internal variability on the changes in circulation and moisture transport, but the comparison with decadal variability indicate that the SIC reduction plays a role here too. The sentence has been revised to "*As a result, decreases in precipitation in the "low" SIC case over the King Haakon VII Sea and Wilkes Land sector can be traced to the reduction in meridional flow and related moisture fluxes from the north due to the SIC decrease and internal variability*"

P10 L22-24 « Although the experimental design in this study doesn't allow us to pinpoint a causal relationship among the three effects (i.e., lower SIC, reduced meridional moisture flux, and precipitation decrease) »

Precipitation decrease is a result of circulation change.

The sentence doesn't add much here, so it has been removed.

P10 L29-31 « Therefore, the impact of sea ice anomalies and corresponding SST changes on

Antarctic precipitation stem both from their direct impact on moisture sources and from the circulation changes that accompany the different SIC and SST patterns »

Your aim is to disentangle SIC change from circulation change. Here SIC does not change much in DJF, so changes in circulation cannot be attributed to changes in SIC.

*As we discussed earlier, the circulation changes cannot be quantitatively attributed to SIC/SST or the internal variability. The sentence has been revised to "Therefore, the impact of sea ice anomalies and corresponding SST changes on Antarctic precipitation stem both from their direct impact on moisture sources and from the circulation changes that accompany the different SIC/SST patterns and can be due in part to interval variability."*

P11 L11-13 « Conversely, the strength and location of the ABSL can also be affected by the sea ice and temperature changes, as depicted in Fig. 10e. »

For JJA only, and significance must be quantified vs. internal variability.

*The significance has been tested against the decadal variability for DJF and JJA. In JJA, the reduced SLP over the ABSL region is larger than the computed decadal variability.*

P11 L18-20 « In this study, we use a coupled atmosphere-land version of the Community Earth System Model (CESM1-CAM5) with explicit water tagging capability to quantify the impact of sea surface temperature (SST) and sea ice concentration (SIC) changes on the moisture sources of Antarctic precipitation. »

You use CAM5 with water tagging, not the coupled model. This sentence is confusing.

*We mean to say that the atmosphere and land are coupled. It has been revised to avoid confusion.*

P11 L23 « 1800 » Typo? 1000?

*Corrected.*

P11 L25-26 « are used as prescribed boundary conditions for atmosphere- only simulations. » Add the length of simulations: 10 years

*Done.*

P11 L28-30 « Because of the prescribed changes in the SIC and SST, surface sensible heat fluxes and evaporation over the lower SIC areas in the Southern Ocean (SO) have a large increase in the "low" SIC case, compared to 30 the "high" SIC case. »

The relation with circulation change must be clarified here, significance of change must be quantified, depending of the season.

*The significance has been tested against the decadal variability. It is indeed more significant in JJA. The sentence is revised accordingly.*

P12 L3-5 « The three remote source regions have a reduced absolute contribution to water vapor further inland in the "low" SIC case, which leads to a discernable reduction in their fractional contribution, especially, in the lower and mid troposphere. »

Because of change in circulation

Not entirely. Changes in evaporation also contribute to the moisture flux into Antarctica.

P12 L7-8 « This is qualitatively consistent with the source attribution change in response to warming from CO2 doubling (Singh et al., 2017). » Explain more.

It is in terms of the annual mean. Seasonal source attribution could be different between the two studies. It has been clarified.

P12 L9-12 « This difference is larger than the interannual variability of Antarctic precipitation (characterized by one standard deviation of annual mean precipitation) within the 10 years of the "mean" SIC case as well as over 1000 years of the CESM LENS experiment. »
And compared to decadal variability?

We didn't calculate the decadal variability of precipitation for the entire Antarctica. However, spatial distributions (by season) are compared in Figs. S6-S10.

P12L32-P13L « The resultant changes in meridional moisture fluxes from the Southern Ocean to the Antarctic continent can intuitively explain some of the precipitation differences between the "low" and "high" SIC cases. » Not quantified, very approximative

Agreed. We didn't mean to go into the quantitative details, especially, in the summary here.

**Figures**

Figure 1
Don't use a divergent colorbar for sea ice concentration. Use a sequential colorbar instead.

The diverging color bar is kept for the difference plots, but sequential colors are now used for the mean SIC.

Figure 2
Don't use a divergent colorbar for evaporation/sublimation, or around 0. Display evaporation in kg m-2 year-1 (= mm year-1)

The top panel has been removed as suggested by Referee #2.

Figure 3
Use symmetric colorscales around 0 (for Fsh, E, and P) Change unit for E and P: kg m-2 year-1

Color scales and units are changed as suggested.

Figure 4
Add a) and b) and change the main text accordingly
For precipitation, give the value in Gt year-1 or in Gt month-1, as in the text.

Changed as suggested.

Figure 5, 6 and 7
Don't use divergent colorbars.

Changed as suggested.

Figure 10
Use symmetric colorscales around 0.

Changed as suggested.

Same remarks for supplement

Changed as suggested.

**Harald Sodemann (Referee #2)**

Review of "Influence of Sea Ice Anomalies on Antarctic Precipitation Using Source Attribution" by Wang et al., submitted to The Cryosphere.

The authors present a sensitivity study of the global water cycle, testing the response of precipitation origin over Antarctica to combined sea ice cover and sea surface temperature changes, using a climate model capable of water vapour tagging. The results are interesting and novel, and fit well into the scope of TC. However, some aspects need further clarification, concerning the more general implications of the findings and the relation to previous results. In addition, the presentation quality of some figures can be improved, as detailed below.

**Major comments**

1. The findings should be placed more clearly into the wider scientific context to make their significance more obvious. This concerns the abstract, introduction and conclusions.

   In response to some comments from referee #1 and the specific comments below, we have revised the manuscript to include a wider context and more comparisons to previous studies (e.g., Krinner et al., 2014; Frieler et al, 2015; Kittel et al., 2018; …).

2. The relation to previous studies should be more clearly specified. The present study is based on a long control run, whereas previous studies mostly used reanalysis data. What are the differences? How valid is the control run for the present-day simulation? And, more specifically, how do the uppermost and lowermost percentiles used here compare to observed natural variability?

   Previous studies using reanalysis data mostly also consider climate change signals and more realistic atmospheric circulations for a specific time period but don't allow **interactive** dynamical feedback from oceanic changes to the atmosphere. To some extent, the prescribed sea ice and SST are disconnected from the atmospheric response (based on the reanalysis data).

   By design our experiments are conducted for the purpose of sensitivity analysis. The oceanic conditions (sea ice concentration and sea surface temperature) are based on a long CESM pre-industrial control run, which enable assessment of unforced internally generated climate variability. The prescribed sea ice anomalies may not be representative of present-day natural variability that is challenging to quantify from short-term observations in the presence of anthropogenic forcing. However, in some regions the pre-industrial sea ice anomalies are comparable to observations during the recent decades (Hobbs et al., 2016). In response to comments from the Referee #1, we have now included a comparison of baseline simulations to ERA5 reanalysis and tested the significance of the model sensitivity to SIC/SST changes against decadal variability. Please see also our responses to Referee #1.

3. The tagging setup should to be modified, or justified more clearly, and be documented more comprehensively. The Antarctic land mass is currently part of a general land tracer in the tagging setup. It would be interesting to specify an Antarctic-land only tracer, to allow for distinguishing local from remote contributions. In addition, the Southern Ocean tag appears

specified in a confusing way, with a narrow stripe going around the globe, and some boxes for the Weddell Sea and others placed inside. It should be stated more clearly why this specific setup has been chosen. Furthermore, a table or other form of description of the exact box coordinates are needed in order to compare and reproduce results.

The total contribution to Antarctic precipitation from global land is small (less than 5%). The lower tropospheric moisture over Antarctica attributed to the land tag is predominantly from the local continent, and the remote contributions from lower latitude land are mostly in the upper troposphere (Fig. 6). The difference in land contribution due to sea ice anomalies is seen in the lower troposphere (Fig. 8). On the other hand, since the focus of this study is on ocean and sea ice, we didn't want to add more land tags to increase the computational burden. Regardless, we agree with the referee that this design is not ideal.

The tagged regions are defined by latitude-longitude coordinates in the model. Smaller regions are used for the Southern Ocean because it is in close proximity to Antarctica and the surface evaporation is more affected by sea ice variations. Therefore, we use regular latitude-longitude boxes to define the Southern Ocean and five some source regions. The remaining area (irregular shape) of the Southern Ocean is constructed by differencing between the entire Southern Ocean tag and the sum of the five regular boxes. The irregular stripe is not a focus of our analysis. The tagged regions are independent of each other. They can have overlaps without an issue of double counting. We have now included a table (Table S1; see below) the describe the tagged regions in the Supplement.

Table S1: The latitude-longitude coordinates for the tagged water source regions. Land mask (and land fraction for coastal areas) in the model is used to define the "land" tag and mask land in the oceanic boxes.

| Source region | Latitude S | Latitude N | Longitude W | Longitude E |
|---|---|---|---|---|
| Land | -90 | 90 | 0 | 360 |
| Subtropical N. Pacific | 10 | 30 | 105 | 260 |
| Gulf of Mexico | 10 | 30 | 260 | 300 |
| Subtropical N. Atlantic | 10 | 30 | 300 | 360 |
| Northern Indian Ocean | 10 | 30 | 35 | 105 |
| Pacific Warm Pool | -10 | 10 | 25 | 190 |
| Equatorial Pacific | -10 | 10 | 190 | 285 |
| Equatorial Atlantic | -10 | 10 | 290 | 25 |
| Southern Indian Ocean | -50 | -10 | 25 | 130 |
| South Pacific | -50 | -10 | 130 | 290 |
| South Atlantic | -50 | -10 | 290 | 25 |
| Southern Ocean | -90 | -50 | 0 | 360 |
| Amundsen Sea | -90 | -60 | 210 | 285 |
| Cosmonauts Sea | -70 | -53 | 30 | 60 |
| Mawson Sea | -90 | -55 | 90 | 120 |
| Weddell Sea | -90 | -55 | 285 | 360 |
| Ross Sea | -90 | -55 | 120 | 210 |

4. The findings in some figures should be condensed further to facilitate grasping the main findings, as detailed in the specific comments

Please see our responses to the specific comments below.

**Specific comments**

**Abstract**

Pg. 2, L. 1: highlight the relevance to other research, now it is just described as a sensitivity study in the first sentence.

It is indeed a sensitivity study but with more realistic sea ice and SST anomalies (in terms of unforced internal variability), compared to previous studies (e.g., Kittel et al., 2018) with prescribed homogeneous perturbations or SIC/SST from different CMIP5 models. More detailed context is provided in the introduction.

Pg. 2, L. 6: "but": consider splitting into two sentences

Done.

Pg. 2, L. 11: A percent number could be more informative here

Added the percent change.

**Introduction**

Pg. 3, L. 1: "SMB ... plays a role... by supplying" does not make sense

Revised.

Pg. 3, L. 10-13: Not sure the exact mechanism is assessed quantitatively in this study. May need rephrasing.

Rephrased.

Pg. 3, L. 17: "Because of..." ultimate meaning/logic of this sentence not clear. "Relies" for what purpose? Maybe it helps to rephrase in terms of mean temperature?

Revised. It means to highlight the importance of moisture transport to local precipitation over Antarctica.

Pg. 3, L. 20: The relevance of knowing the moisture source could be highlighted here.

Yes, it is highlighted at the end of the paragraph.

Pg. 3, L. 24 onward: Some link to the state of the Antarctic hydrologic system from observation or reanalysis data would be useful here (e.g., Papritz et al., 2014).

Thanks for the suggestion. The following two sentences have been added. *Sodemann and Stohl (2009) showed that the source regions for Antarctic precipitation over the SO vary greatly between the ocean basins. Based on reanalysis datasets Papritz et al. (2014) found that extratropical cyclones and fronts are key to the spatial distribution of evaporation and*

*precipitation over the SO as well as moisture fluxes toward Antarctica.*

Pg. 4, L. 1: Clarify the distinction between internal and natural climate variability

Both "natural" and "internal" refer to "unforced" climate variability. To avoid confusion, it has been changed to "internal climate variability".

Pg. 4, L. 2: "such responses": unclear what exactly is referred to

It is referred to the sea ice changes in the past few decades. This has now been clarified.

Pg. 4, L. 10: "and/or" - rephrase

Done.

Pg. 4, L. 17: "unacceptably" - I think this depends on the approach and purpose. One frequently used countermeasure is to consider the problem stochastically, by calculating many trajectories, as is done in Sodemann and Stohl (2009). Interpreting single trajectories beyond 10 days in contrast may be meaningless. On the other hand, tracer studies suffer from numerical diffusion and the uncertainty of parameterisation processes, and do not provide the spatial detail of source contributions available from backward trajectories. A more balanced discussion would be justified here.

Done.

Pg. 4, L. 20 onward: The relevance for ice core studies could be included as an additional motivation, see for example Winkler et al., 2012, Wang et al., 2013, Masson- Delmotte et al., 2011, Buizert et al., 2018 and references therein.

Added. Both back-trajectory and GCM water tracer approaches along with ice core records of water isotopic composition have been used to attribute water sources at Antarctic ice core sites and study their historical changes (e.g., Masson- Delmotte et al., 2011; Wang et al., 2013, Buizert et al., 2018).

Pg. 4, L. 22: "such studies" - which ones specifically?

Referred to the aforementioned back-trajectory and water tracer studies. This has now been clarified.

Methods

Pg. 5, L. 3: Are all 5 references needed as reference to the method, or rather example applications? Please clarify.

These are mostly example applications. This has now been clarified.

Pg. 6, L. 3: "assuming" - has it been checked that the simulation stabilises after 1 year?

Yes, it has been checked. With the prescribed SST and sea ice, the atmospheric model stabilizes within a year.

Pg. 6, L. 13: It would be helpful to more convincingly illustrate and quantify this aspect.

Figures are now shown in Figs. S1-S2.

Pg. 6, L. 23: tags -> tag

Corrected.

Pg. 6, L. 25: remove "well"

Done.

Pg. 6, L. 26: "differencing": rephrase, e.g. distinguishing?

Rephrased to refer to the difference between the Southern Ocean and the sum of the five sub-regions.

Pg. 6, L. 15 onward: SST mean and anomalies should be shown/discussed and placed in relation to observations/reanalysis.

Comparison to ERA5 reanalysis are now included in Figs. S1-S4.

Pg. 6, L. 15: see major comment 3.

Revised accordingly.

**Results**

Pg. 7, L. 11: "echoing the finding" - what aspect specifically?

The positive correlation between moisture convergence and precipitation, as described in the preceding sentence. Revised to "echoing the same finding of…".

Pg. 7, L. 20: This number could be useful to include in the Abstract

Added.

Pg. 7, L. 22 onward: would be helpful to put these numbers into the context of observation /reanalysis data

It's a good idea, but the comparison could be misleading because different reanalysis products give quite different precipitation results according to Bromwich et al. (2011).

Bromwich, D. H., Nicolas, J. P., and Monaghan, A. J.: An Assessment of Precipitation Changes over Antarctica and the Southern Ocean since 1989 in Contemporary Global Reanalyses. J. Climate, 24, 4189–4209, https://doi.org/10.1175/2011JCLI4074.1, 2011.

Pg. 7, L. 27: what is meant by "more significant"?

The sentence has been revised to "The contrast in Antarctic precipitation contributed by S. Ocean between the "low" and "high" SIC cases, 102 Gt year$^{-1}$, is much larger than the interannual variability of 35 Gt year$^{-1}$ in precipitation that originates from the S. Ocean…".

Pg. 8, L. 1: It would be interesting to know what land region contributes, in particular anything from Antarctica.

Please see the response to major comment #3.

Pg. 8, L, 2 onward: it would be helpful to collect these results in a table.

The manuscript and Supplement are already quite long.  We don't see a need to add another table to repeat the information.

Pg. 8, L. 8: The discussion could be more comprehensive. My take is that the overall results are quite similar, and specific numbers depend on how the ocean sectors have been defined here and in Sodemann and Stohl (2009). That itself is a useful finding that should be stated clearly. Part of the differences may then be due to the fact that Sodemann and Stohl (2009) based their results on ECMWF analyses for a specific period, while you consider a control run of a climate model. Comparison to reanalysis data may therefore be helpful to better explain the differences found here. Furthermore, a table or other form of description of the exact box coordinates are needed in order to compare (and reproduce) results.

We agree that the quantitative difference in the annual mean contribution from results of Sodemann and Stohl (2009) based on reanalysis for a specific time period may be due to internal variability (as opposed to the source attribution tools). The seasonal cycle of the S. Ocean contribution might be due to the sea ice or circulation difference between the two models. We have revised the text to reflect this.

A table (Table S1) describing the exact latitude-longitude coordinates has been added to the Supplement.

Pg. 8, L. 30: Fig S4 seems to contain useful results but the information needs to be condensed more (see below).

Addressed below (Figures)

Pg. 9, L. 9: See, for example, Stohl and Sodemann (2010), which also clearly illustrates the thermodynamic transport barrier to low-level airmasses from the Southern Ocean (their Fig. 3).

Added.

Pg. 9, L. 9: "at the source" - or underways!

Revised.

Pg. 9, L. 17: The presentation of the results can be improved, see comments on figures below.

Addressed below (Figures)

Pg. 9, L. 25 onward: The discussion here is quite vague, and could be made more specific

Revised.

Pg. 10, L. 4 onward: What is the region for the apparently substantial changes in the NH? Maybe better to only show the SH.

Those are regions with differences in SST. Figures are revised to only show NH since the SIC change is the focus.

Pg. 10, L. 30: It appears you imply a direct and indirect impact from the SIC and SST anomalies - if this is a main result, this should be introduced more prominently already in the Introduction. Is it really possible to separate both aspects, as dynamic changes can also affect evaporation of the source regions?

The current results suggest that both evaporation and dynamical feedback have an impact on the difference in moisture source attribution between the low and high SIC cases. However, we agree with the referee that it is impossible to clearly separate the effect of evaporation and dynamical feedback in the current simulations. Therefore, we choose to illustrate the changes in moisture flux and transport but cannot quantify their relative contributions, not even mentioning

the higher order effect (e.g., thermodynamic and dynamic feedback on evaporation).

Pg. 11, L. 15: The takeaway from this paragraph is not fully clear.

This is basically to provide a context of the complicated circulation changes and dynamic feedback from the SIC/SST anomalies, which could have affected the moisture transport to Antarctica. We admit their existence but cannot separate the impact from the direct effect of SO evaporation on Antarctic precipitation. A future study with more carefully designed series of experiments (e.g., with specified large-scale circulations, surface wind stress, evaporation and heat fluxes) is needed to address this. We have added a sentence to clarify.

**Summary**

Pg. 12, L. 15: It could be helpful to state whether or not this confirms earlier findings (to my understanding it does)

Done.

Pg. 12, L. 34: "intuitively" - may not be applicable to all readers

Removed.

Pg. 13, L. 1-5: Would be useful to highlight the wider implications of this study in the end.

Added.

**Figures**

Fig. 1: In Antarctica, seasons are more commonly defined as JFM and ASO, in line with the sea ice seasonality. Maybe it would probably be sufficient to show these seasons only along with the annual mean.

We have taken the suggestion to only show austral summer (DJF) and winter (JJA) along with the annual mean. DJF and JJA are still preferred for the consistency with other results that consider seasonality in lower latitudes.

Fig. 2: The E panel does not appear to be relevant here, but could be part of a "validation" figure where you compare SH plots of annual mean P and E from the model simulation with reanalysis/observation data.

We have removed the top panel for evaporation. Both E and P are now evaluated against ERA5 reanalysis in Figs. S1 and S2.

Fig. 4: This figure could be made more easily readable by either showing bars for 3- month periods, or by removing the white space in between the individual monthly bars. Similarly, Fig. S4 could be made into a much simpler figure that only compares the annual mean or summer/winter differences for the 3 regions.

We appreciate the thoughtful suggestions. The plots are a little complicated but still readable, and we do like to keep the monthly information. Nonetheless, we have reduced much of the white spaces in between the color bars.

Fig. 5: Panel a is method validation and could be removed in this context, the big red spot draws

a lot of attention, and prevents a more useful color scale to be used. For the purpose of the paper, it seems it would be more useful to highlight the contributions to precipitation in Antarctica only, by masking the tracer contribution over the Oceanic regions, and zooming in on Antarctica.

Panel (a), which is not the sum of all source regions, does make a point that the five major source regions together account for over 95% of total Antarctic precipitation. We have taken the suggestion to zoom in more to highlight Antarctica.

Fig. 6: Instead of a zonal mean covering all latitudes, it would be more useful to highlight the Southern Hemisphere only. In fact, Fig. 6 may be dropped altogether, and be replaced by Fig. S5.

Revised.

Fig. 7: Similarly, the figure would be improved by showing the SH part only.

Changed as suggested.

Fig. 8: May be dropped in favour of Fig. 9

Figure 8 has been moved to the Supplement and Figure 9 is kept in the manuscript.

Fig. 9: Show SH section only (or discuss the NH changes which are in general larger or as large as the NH changes)

Changed as suggested.

Fig. 10: Busy figure with 15 panels and the SLP difference contours. Consider removing panel c and d, and e, or keeping e and removing the contours from the other panels.

Changed as suggested. The original Fig. 10 has been moved up, becoming the new Fig. 4, as suggested by Referee #1.

References

Buizert, Christo; Sigl, Michael; Severi, Mirko; Markle, Bradley; Wettstein, Justin; Mc- Connell, J; Pedro, Joel; Sodemann, H.; Goto-Azuma, Kumiko; Kawamura, Kenji; Fujita, Shuji; Motoyama, Hideaki; Hirabayashi, Motohiro; Uemura, Ryu; Stenni, Barbara; Par- renin, Frederic; He, Feng; Fudge, T.J.; Steig, Eric J., 2018: Abrupt ice-age shifts in southern westerly winds and Antarctic climate forced from the north, Nature 563: 681- 685.

Masson-Delmotte, V., Buiron, D., Ekaykin, A., Frezzotti, M., Gallée, H., Jouzel, J., Krin- ner, G., Landais, A., Motoyama, A., Oerter, H., Pol, K., Pollard, D., Ritz, C., Schlosser, E., Sime, L. C., Sodemann, H., Stenni, B., Uemura R., and Vimeux, F., 2011: A comparison of the present and last interglacial periods in six Antarctic ice cores Clim. Past, 7, 397-423.

Papritz, L., Pfahl, S., Rudeva, I., Simmonds, I., Sodemann, H., and Wernli, H., 2014: The role of extratropical cyclones and fronts for Southern Ocean freshwater fluxes, J. Climate 27: 6205– 6224, doi:10.1175/JCLI-D-13-00409.1.

Stohl, A., and Sodemann, H., 2010: Characteristics of atmospheric transport into the Antarctic troposphere, J. Geophys. Res., 115, D02305, doi:10.1029/2009JD012536.

Wang, Y., Sodemann, H., Hou, S., Masson-Delmotte, V. Jouzel, J. and Pang, H., 2013: Snow accumulation and its moisture origin over Dome Argus, Antarctica. Clim. Dyn., 40:731-742, doi: 10.1007/s00382-012-1398-9.

Winkler, R., Landais, A., Sodemann, H., Dümbgen, L., Priéa, F., Masson-Delmotte, V., Stenni, B., and Jouzel, J., 2012: Deglaciation records of 17O-excess in East Antarctica: reliable reconstruction of oceanic relative humidity from coastal sites. Clim. Past 8, 1- 16.

**Anonymous Referee #3**

General Comments:

As far as it goes, this exploration of the impact of sea ice extent anomalies on Antarctic precipitation within the climate model context is competently done and the findings are interesting and valuable. Unfortunately, there are many things left dangling that require some effort to rectify before publication.

1. This is a model study and the title should reflect this.

The title has been changed to "Influence of sea ice anomalies on Antarctic precipitation using source attribution in the Community Earth System Model"

2. There is very little effort made to relate the results to the real world, rather the manuscript seems to assume that the results must be realistic. What constraints can you apply throughout the manuscript to the results to verify their credibility? In the background, stable water isotope studies are relevant, so it would be nice to see explicit discussion of relevant results.

We did compare our results with the CESM LENS pre-industrial control simulation. We have also considered suggestions from the other two referees to compare some of our results to ERA5 reanalysis products. Please see Figs. S1-S4 and relevant responses to their comments.

3. Explain why you used a pre-industrial control as the basis for your atmospheric sensitivity studies. What difference does this make to today? Do you think that fixed SST and sea ice distort your results in contrast to having an interactive ocean?

The model experiments were designed to isolate the impact of Southern Ocean SIC/SST anomalies on source–receptor relationships for moisture and precipitation over Antarctica from the anthropogenic warming (e.g., associated with GHGs). Thus we took the SIC and SST anomalies from the pre-industrial control simulation of CESM Large Ensemble simulations (LENS), which enable assessment of unforced internally generated climate variability. The prescribed SIC/SST anomalies may not be representative of present-day natural variability that is challenging to quantify from short-term observations in the presence of anthropogenic forcing.

Kittel et al. (2018) did similar sensitivity experiments using homogeneously perturbed SIC or SIC/SST from different CMIP5 models. We are using more realistic SIC/SST anomalies (in terms of unforced internal variability in the same model). With an interactive ocean, the model would take a long time to stabilize and generate the internal variability in SIC/SST or, alternatively, by introducing a strong external forcing (e.g., $2\times CO_2$) to attain the desired change in SIC/SST. It is almost undoable with the many water tracers enabled in the simulations.

4. There are some unexplained results for low versus high sea ice. What is the reason large PW increase north of 55S between 90E and 120E (Fig. 3c)? Why does the surface sensible heat flux decrease north of 55s (Fig. 3d)? Why does the latent heat flux decrease north of 55S between 90E and 170E (Fig. 3e)? This impacts the precipitation (Fig. 3f). You could discuss/explain these results after presenting Fig. 10.

These are all good questions. In addition to evaporation and precipitation, an important factor that affect the column-integrated water (PW) is the moisture convergence/divergence. For the specific location, the increase in PW appears to be consistent with the reduction in northward

meridional moisture fluxes (new Fig. 4a). The large differences in sensible heat flux and evaporation over the northern latitudes of the SO between the two cases are due to meteorological responses (e.g., wind speed, temperature, and humidity) to the SIC/SST anomalies. For example, the decrease of wind speed (north of 55S) is clearly shown in Fig. S5. Following the suggestion, we have now discussed more along with the Fig. 10 (new Fig. 4).

5. You use unusual units for P, g/m*m/h, in contrast to the frequent mm/d or mm/yr. The latter units allow the reader to evaluate the magnitude of the simulated changes.

In response to a similar comment from Referee #1, units for P and E have been changed to kg m$^{-2}$ year$^{-1}$ that are equivalent to mm year$^{-1}$.

Smaller Comments.

6. Page 3, line 17: Palerme et al. (2016) as per reference list?

Corrected it in the reference list.

7. Page 4, line: Difference between natural and internal climate variability?

Both "natural" and "internal" refer to "unforced" climate variability. To avoid confusion, it has been changed to "internal climate variability".

8. Page 5, line 2: Need Hurrell et al. (2013) reference.

Added to the reference list.

9. Page 6, line 8: "further" than what?

It's not being used as a comparative adverb. It can be removed.

10.  Page 6, line 10: What is "CESM LENS"?

Now defined.

11. Page 7, line 11: Must be "anomalous meridional moisture transport divergence" to fit with the atmospheric water balance equation.

Yes, it is meant to be "meridional moisture flux divergence" term in the water budget equation. It has been corrected.

12. Page 7, line 18: Fall and spring are when the low-pressure trough around Antarctica is closest to the continent, known as the semi-annual oscillation.

Thanks for pointing this out.

13. Page 8, line 1: "Evaporation/sublimation over land". This is a quite surprising result. Do you mean primarily over Antarctica in summer?

It does have a peak in austral summer (December and January, see new Fig. 5) but we cannot tell whether it is primarily over Antarctica since the 'land' tag represents the global land.

14. Page 8, line 24: Why do you think that the remote sources mostly lead to precipitation decreases?

We didn't mean to suggest that the remote sources mostly lead to Antarctic precipitation decreases. The annual mean Antarctic precipitation is 150 Gt year$^{-1}$ more in the low SIC case

than in the high SIC case, among which 102 Gt year$^{-1}$ is explained by the difference in Southern Ocean contributions and thus less by the remote sources. Therefore, there is indeed a decrease in the fractional contribution by the remote sources in the low SIC case relative to the high SIC case. The sentence has been revised to avoid confusion: *"This arises because small increases in precipitation originating from remote sources can be overwhelmed by large increases from local sources."*

15. Figs. 3, 8-10, S2, S3: Statistical significance should be tested for these figures.

Done.

16. Fig. 10f: Please use the more physically meaningful hPa rather than Pa for SLP differences.

Changed as suggested.

---

## Author Response (AR2)

We thank the referees for their careful review and thoughtful comments. Below please see our point-by-point responses and changes made to the manuscript.

**Referee report #1**

Thank you for responding to my comments. It is now clearer what is going on, but some additional enhancements are desirable.

You need to state clearly in the abstract and probably in the manuscript as well that you are using pre-industrial conditions to examine sea-ice generated Antarctic precipitation variability in the absence of anthropogenic forcing. This should appear explicitly in the first sentence of the abstract, I think, rather than being implicit here and throughout the manuscript. I am recommending minor revisions, but think that this is a very important aspect to address comprehensively to make the intent and methodology of your extensive work obvious to the reader.

Done as suggested. Now the first sentence of the abstract reads "We conduct sensitivity experiments using a general circulation model that has an explicit water source tagging capability forced by prescribed composites of pre-industrial sea ice concentrations (SIC) and corresponding sea surface temperatures (SST) to understand the impact of sea ice anomalies on regional evaporation, moisture transport, and source–receptor relationships for Antarctic precipitation in the absence of anthropogenic forcing." A similar statement is also made in the summary paragraph of the Introduction section: "In this study, we aim to understand the impact of SO sea ice anomalies associated with internal variability (in the absence of anthropogenic forcing) on local evaporation, moisture transport and source–receptor relationships for moisture and precipitation over Antarctica using a GCM that has an explicit water source tagging capability."

Section 3.2: You attribute southerly katabatic flow to the polar high. Actually, the katabatic winds are caused by the radiative cooling of surface air over the continent along with the force of gravity acting on the cold air over sloping terrain. The Earth's rotation via the Coriolis effect also exerts a significant impact.

Although the annual mean Antarctic surface winds are primarily katabatic in origin, the katabatic flow was loosely used here to describe the vertically integrated southerly outflow. We agree with the referee about the confusion here. We have now removed the phrase "associated with the polar high".

**Referee report #2**

Accepted as is. No further comment.

**Referee report #3**

The authors answered to all my comment and I acknowledge the large amount of work they did for improving their manuscript. I recommend this article to be published in The Cryosphere after addressing the following minor suggestions:

With regards to Fig. S1-S4, I suggest to replace the 4 figures by 2 figures showing the differences between the variables (baseline simulation - ERA5), with hashes where significantly different, as for new Figures 3 and 4. A focus on large scale circulation variables only is sufficient, as ERA5 does not assimilate Precipitation and Evaporation.

Following the referee's suggestion, we have made a new figure to show the difference (i.e., baseline simulation minus ERA5) for the corresponding large-scale circulation variables plotted in the original Figs. S3 and S4. The new figure, which is also shown below (Fig. R1), now becomes Fig. S4. Since the baseline simulation (10 years) results and the ERA5 (40 years) reanalysis have different number of years, it is challenging to make a meaningful statistical significance test. Instead, we use stippling to mark areas having the difference larger than the decade variability derived from the 1100-year CESM-LENS simulations (shown in Fig. S11). We have decided to also keep the original figures (i.e., Figs. S1, S2 and S3) that can provide additional information of the baseline simulation and ERA5 reanalysis.

Change the following paragraph accordingly: « Although these sensitivity simulations are not designed to represent present-day conditions, several essential model fields from the baseline simulation are compared to the fifth generation ECMWF reanalysis (ERA5, 1979-2018). The main purpose is to provide a context for the interpretation of model results that might also be valid for the recent historical period in terms of internal climate variability. The large-scale patterns of SIC, surface temperature, circulation (sea level pressure), precipitation, precipitable water, and horizontal moisture fluxes in the baseline simulation are comparable to those in the ERA5 reanalysis, as shown in Figs. S1-S4. »

**The text has been revised accordingly.**

With regard to the answer to my comment regarding SIC and SST, I do believe that it is a methodology error to take the mean of multiple models instead of the median. In the answer: « Certainly, SIC in the ERA5 reanalysis reflects anthropogenic forcing already. Surface temperature differences are consistent with SIC differences. We don't see any unrealistic boundary conditions that the referee concerned about. »

Of course by comparing averaged SIC over several years you cannot see the impact of averaging the SIC throughout several simulation. If you do the same for 1 month in ERA5 vs. 1 month in the baseline simulation, then you will see a huge difference of spatial patterns.

I do not ask for you to re-do the simulation, but at least to acknowledge that it might not be the best way to proceed, maybe for future work.

We appreciate the referee's further explanation of this concern, and we acknowledge that the way of averaging the SIC over many years might not be the best way to represent short-term mean sea ice spatial distribution. We will keep this in mind for future work.

« Comparison to the corresponding decadal variability of these annual mean fields (Fig. S6), along with a Student's t-test at 90% confidence, suggests that the significant regional differences in surface temperature, evaporation and precipitable water are mostly due to SIC/SST perturbations while changes in precipitation is influenced more by internal variability. »

Why do you use a Student's t-test and not directly the estimate of decadal variability to hash the differences as significant?

We did look into this way of marking the significance of differences. As shown in the Fig. R2 and Fig. R3, corresponding to Fig. 3 and Fig. 4, respectively, almost all the areas with large differences are significant. This can be easily seen by comparing with the decadal variability distribution plotted in the current Figs. S6 and S11. However, the Student's t-test results can also evaluate the significance of the difference between the two means within the 10 simulation years. A similar test has been done to other difference plots for water source attribution that cannot be obtained from alternative ways. To keep the additional information and consistency, we decide to use the Student's t-test results to mark the significance in Figs. 3 and 4, while the comparison to decadal variability is referred to Figs. S6 and S11 with a description in words.